# Genetic and chemotherapeutic influences on germline hypermutation

Joanna Kaplanis[1], Benjamin Ide[2], Rashesh Sanghvi[1], Matthew Neville[1], Petr Danecek[1], Tim Coorens[1], Elena Prigmore[1], Patrick Short[1], Giuseppe Gallone[1], Jeremy McRae[1], Genomics England Research Consortium*, Jenny Carmichael[3], Angela Barnicoat[4], Helen Firth[1,3], Patrick O'Brien[2], Raheleh Rahbari[1] & Matthew Hurles[1✉]

Mutations in the germline generates all evolutionary genetic variation and is a cause of genetic disease. Parental age is the primary determinant of the number of new germline mutations in an individual's genome[1,2]. Here we analysed the genome-wide sequences of 21,879 families with rare genetic diseases and identified 12 individuals with a hypermutated genome with between two and seven times more de novo single-nucleotide variants than expected. In most families (9 out of 12), the excess mutations came from the father. Two families had genetic drivers of germline hypermutation, with fathers carrying damaging genetic variation in DNA-repair genes. For five of the families, paternal exposure to chemotherapeutic agents before conception was probably a key driver of hypermutation. Our results suggest that the germline is well protected from mutagenic effects, hypermutation is rare, the number of excess mutations is relatively modest and most individuals with a hypermutated genome will not have a genetic disease.

The average number of de novo mutations (DNMs) generating single-nucleotide variants (SNVs) is estimated to be 60–70 per human genome per generation, but little is known about individuals with germline hypermutation with unusually large numbers of DNMs[1,3,4]. The human germline-mutation rate varies between individuals, families and populations, and has evolved over time[5–9]. Parental age explains a large proportion of variance for SNVs, insertion–deletions (indels) and short tandem repeats[1,10,11] It has been estimated that there is an increase of around 2 DNMs for every additional year in father's age and around 0.5 DNMs for every additional year in mother's age[1,12]. Subtle differences have also been observed between the maternal and paternal mutational spectra and may be indicative of different mutagenic processes[2,13–15]. Different mutational mechanisms can leave distinct mutational patterns termed 'mutational signatures'[16,17]. There are currently more than 100 somatic mutational signatures that have been identified across a wide variety of cancers of which half have been attributed to endogenous mutagenic processes or specific mutagens[18,19]. The majority of germline mutations can be explained by two of these signatures, termed signature 1 (SBS1), probably due to deamination of 5-methylcytosine[20], and signature 5 (SBS5), which is thought to be a pervasive and relatively clock-like endogenous process. Both signatures are ubiquitous among normal and cancer cell types[21,22] and have been reported previously in trio studies[14]. The impact of environmental mutagens has been well established in the soma but is not as well understood in the germline[23,24]. Environmental exposures in parents, such as ionizing radiation, can influence the number of mutations transmitted to offspring[25–27]. Individual mutation rates can also be influenced by genetic background. With regard to somatic mutation, thousands of inherited

germline variants have been shown to increase cancer risk[28–30]. Many of these variants are in genes that encode components of DNA-repair pathways which, when impaired, lead to an increase in the number of somatic mutations. However, it is unclear whether variants in known somatic mutator genes can influence germline-mutation rates. There are examples in which the genetic background has been shown to affect the local germline-mutation rate of short tandem repeats, minisatellites and translocations[31–35].

An increasing germline-mutation rate results in an increased risk of offspring being born with a dominant genetic disorder[36]. Long-term effects of mutation rate differences as a result of mutation accumulation have been demonstrated in mice to have effects on reproduction and survival rates and there may be a similar impact in humans[37,38].

Little is known about rare outliers with extreme mutation rates. DNMs are a substantial cause of rare genetic disorders and cohorts of patients with such disorders are more likely to include individuals with germline hypermutation[12,39]. To this end, we sought to identify individuals with germline hypermutation in sequenced parent–offspring trios from two rare disease cohorts. We identified genetic or environmental causes of this hypermutation and estimated how much variation in the germline-mutation rate that this may explain.

## Individuals with germline hypermutation

We identified individuals with germline hypermutation in two separate cohorts comprising parent–offspring trios: 7,930 exome-sequenced trios from the Deciphering Developmental Disorders (DDD) study and 13,949 whole-genome sequenced trios in the rare disease arm of the

[1]Wellcome Sanger Institute, Wellcome Genome Campus, Hinxton, UK. [2]Department of Biological Chemistry, University of Michigan, Ann Arbor, MI, USA. [3]East Anglian Medical Genetics Service, Cambridge University Hospitals, Cambridge, UK. [4]North East Thames Regional Genetics Service, Great Ormond Street Hospital, London, UK. *A list of authors and their affiliations appears at the end of the paper. ✉e-mail: meh@sanger.ac.uk

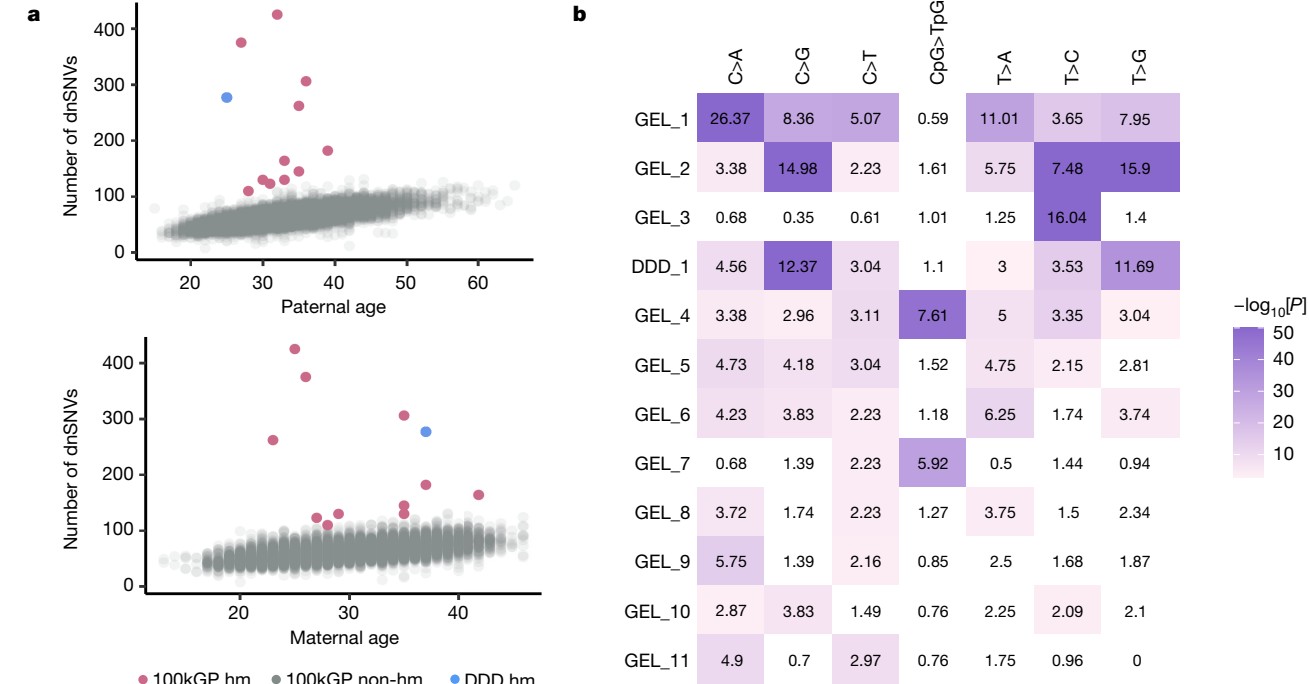

**Fig. 1 | Identification of individuals with germline hypermutation.**
**a**, Paternal and maternal age versus the number of dnSNVs. Individuals with hypermutation (hm) from the 100kGP cohort (pink) and individuals with hypermutation from the DDD cohort (blue) are highlighted. **b**, Enrichment (observed/expected) of mutation type for individuals with hypermutation. Sample names are shown on the *y* axis, and mutation type is shown on the *x* axis.

The enrichment is coloured by the $-\log_{10}$[enrichment *P* value], determined using two-sided Poisson tests comparing the average number of mutations in each type across all individuals in the 100kGP cohort. White colouring indicates no statistically significant enrichment after multiple-testing correction (*P* < 0.05/12 × 7 tests). Exact *P* values are provided in Supplementary Table 2.

100,000 Genome Project (100kGP). We selected nine trios from the DDD study with the largest number of DNMs, given their parental ages, which were subsequently whole-genome sequenced to characterize DNMs genome-wide. In the 100kGP cohort, we performed filtering of the DNMs, which resulted in a total of 903,525 de novo SNVs (dnSNVs) and 72,110 de novo indels (dnIndels). The median number of DNMs per individual was 62 for dnSNVs and 5 for dnIndels (median paternal and maternal ages of 33 and 30) (Supplementary Fig. 1).

We observed an increase in the total number of dnSNVs of 1.28 dnSNVs per year of paternal age (95% confidence interval (CI) = 1.24–1.32, *P* < 10$^{-300}$, negative binomial regression) and an increase of 0.35 dnSNVs per year of maternal age (95% CI = 0.30–0.39, *P* = 3.0 × 10$^{-49}$, negative binomial regression) (Fig. 1a). We phased 241,063 dnSNVs and found that 77% were paternal in origin, in accordance with previous estimates[13–15]. Estimates of the parental age effect in the phased mutations were similar to the unphased results: 1.23 paternal dnSNVs per year of paternal age (95% CI = 1.14–1.32, *P* = 1.6 × 10$^{-158}$) and 0.38 maternal dnSNVs per year of maternal age (95% CI = 0.35–0.41, *P* = 6.6 × 10$^{-120}$) (Extended Data Fig. 1). Paternal and maternal age were also significantly associated with the number of dnIndels: an increase of 0.071 dnIndels per year of paternal age (95% CI = 0.062–0.080, *P* = 8.3 × 10$^{-56}$; Extended Data Fig. 1) and a smaller increase of 0.019 dnIndels per year of maternal age (95% CI = 0.0085–0.029, *P* = 3.4 × 10$^{-4}$; Extended Data Fig. 1). The ratios of paternal to maternal mutation increases per year were very similar—3.7 for SNVs and 3.8 for indels. The proportion of DNMs that phase paternally increased by 0.0017 for every year of paternal age (*P* = 3.37 × 10$^{-38}$, binomial regression; Supplementary Fig. 2). However, the proportion of DNMs that phase paternally in the youngest fathers remains around 0.75 and, therefore, the paternal age effect alone does not fully explain the strong

paternal bias[15]. We compared the mutational spectra of the phased DNMs and found that maternally derived DNMs have a significantly higher proportion of C>T mutations (0.27 maternal versus 0.22 paternal, *P* = 3.24 × 10$^{-80}$, binomial test), whereas paternally derived DNMs have a significantly higher proportion of C>A, T>G and T>C mutations (C>A: 0.08 maternal versus 0.10 paternal, *P* = 4.6 × 10$^{-23}$; T>G: 0.06 versus 0.7, *P* = 6.8 × 10$^{-28}$; T>C: 0.25 versus 0.26, *P* = 1.6 × 10$^{-5}$, binomial test; Extended Data Fig. 2a). These mostly agree with previous studies, although the difference in T>C mutations was not previously significant[13]. Most paternal and maternal mutations could be explained by SBS1 and SBS5, with a slightly higher contribution of SBS1 in paternal mutations (0.16 paternal versus 0.15 maternal, $\chi^2$ test, *P* = 2.0 × 10$^{-5}$; Extended Data Fig. 2b).

We identified 12 individuals with germline hypermutation after accounting for parental age (Methods): 11 from 100kGP and 1 from DDD (Fig. 1a and Extended Data Table 1). The number of dnSNVs for each of the 12 individuals with hypermutation ranged from 110 to 425, corresponding to a fold increase of 1.7–6.5 compared with the median number of dnSNVs per individual. Two of these individuals also had a significantly increased number of dnIndels (Extended Data Table 1). The mutational spectra across these individuals with hypermutation varied considerably (Fig. 1b, Extended Data Figs. 3 and 4 and Supplementary Tables 1 and 2) and, after extracting mutational signatures, we found that, although most mutations mapped onto known somatic signatures from COSMIC[40], a new signature, SBSHYP, was also extracted (Fig. 2, Extended Data Fig. 5 and Supplementary Table 3). In addition to mutational spectra, we evaluated the parental phase, transcriptional strand bias (Extended Data Fig. 6) and the distribution of the variant allele fraction (VAF) for these mutations (Extended Data Fig. 7). After examining these properties, we identified three potential sources of

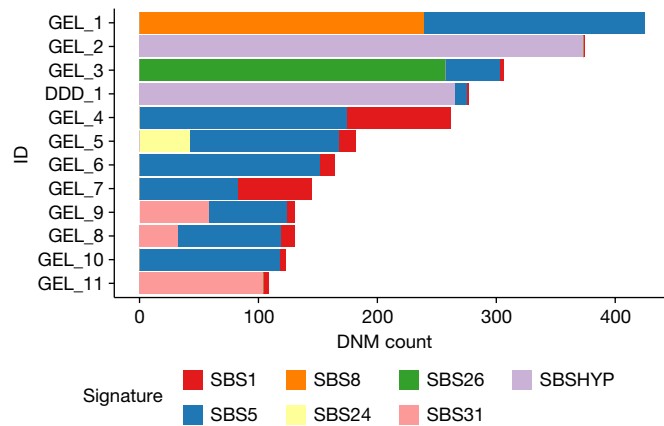

**Fig. 2 | Mutational signatures in individuals with germline hypermutation.** Contributions of mutational signatures extracted using SigProfiler and decomposed onto known somatic mutational signatures as well as the signature SBSHYP that we identified in both DDD_1 and GEL_2.

germline hypermutation: paternal defects in DNA-repair genes, paternal exposure to chemotherapeutics and post-zygotic mutational factors.

## Paternal defects in DNA-repair genes

For eight individuals with hypermutation, the DNMs phased paternally significantly more than expected ($P < 0.05/12$ tests, two-sided binomial test; Extended Data Table 1), implicating the paternal germline as the origin of the hypermutation. Two of these fathers carry rare homozygous non-synonymous variants in known DNA-repair genes (Supplementary Table 4). Defects in DNA repair are known to increase the mutation rate in the soma and may have a similar effect in the germline. Individual GEL_1 has the highest number of DNMs of all of the individuals, and a significantly increased number of dnIndels. The mutational spectra exhibit enrichment of C>A and T>A mutations (Fig. 1b) and we observed a large contribution of the signature SBS8 (Fig. 2). This signature is associated with transcription-coupled nucleotide-excision repair (NER) and typically presents with transcriptional strand bias. This agrees with the strong transcriptional strand bias observed in GEL_1 ($P = 2.1 \times 10^{-40}$, Poisson test; Extended Data Fig. 6). The father has a rare homozygous nonsense variant in the gene *XPC* (Extended Data Table 1 and Supplementary Table 4), which is involved in the early stages of the NER pathway. The paternal variant is annotated as pathogenic for xeroderma pigmentosum in ClinVar and the father had already been diagnosed with this disorder. Patients with xeroderma pigmentosum have a high risk of developing skin cancer and have an increased risk of developing other cancers[41,42]. *XPC* deficiency has been associated with a similar mutational spectrum to the one that we observed in GEL_1 (ref. [43]) and *XPC* deficiency in mice has been shown to increase the germline-mutation rate at two short tandem repeat loci[44].

GEL_3 has about a fivefold enrichment of dnSNVs, which exhibit a distinctive mutational spectrum with around a seventeenfold increase in T>C mutations but no increase in other mutations (Fig. 1b and Extended Data Fig. 3d). Extraction of mutational signatures revealed that the majority of mutations mapped onto SBS26, which has been associated with defective mismatch repair. The father has a rare homozygous missense variant in the gene *MPG* (Extended Data Table 1 and Supplementary Table 5). *MPG* encodes *N*-methylpurine DNA glycosylase (also known as alkyladenine-DNA glycosylase), which is involved in the recognition of base lesions, including alkylated and deaminated purines, and initiation of the base-excision repair pathway. The *MPG* variant is rare in gnomAD (allele frequency = $9.8 \times 10^{-5}$, no observed homozygotes) and is predicted to be pathogenic (CADD score = 27.9) and the amino acid residue is fully conserved across 172 aligned protein sequences from

VarSite[45,46]. The variant amino acid forms part of the substrate-binding pocket and probably affects substrate specificity (Fig. 3a). *MPG* has not yet been described as a cancer-susceptibility gene, but studies in yeast and mice have demonstrated variants in this gene and, specifically, in its substrate-binding pocket, can lead to a mutator phenotype[47,48] (Supplementary Table 6). We examined the functional impact of the observed A135T variant using in vitro assays (Methods and Extended Data Figs. 8 and 9). The A135T variant caused a twofold decrease in excision efficiency of the deamination product hypoxanthine (Hx) in both the T and C contexts (Fig. 3c and Extended Data Fig. 9), with a small increase in excision efficiency of an alkylated adduct 1,*N*(6)-ethenoadenine (εA) in both the T and C contexts (Fig. 3b and Extended Data Fig. 8). The maximal rate of excision is increased by twofold for εA—among the largest increases that have been observed for 15 reported *MPG* variants (Supplementary Table 5). Another variant—N169S, which also shows an increase in *N*-glycosidic bond cleavage with the εA substrate—has been established as a mutator in yeast[48,49]. These assays confirm that the A135T substitution alters the *MPG*-binding pocket and changes the activity towards different DNA adducts. *MPG* acts on a wide variety of DNA adducts and further functional characterization and mechanistic studies are required to link the observed T>C germline mutational signature to the aberrant processing of a specific class of DNA adducts.

## Parental chemotherapy before conception

Three individuals with hypermutation (GEL_8, GEL_9 and GEL_11) have a contribution from the signature SBS31 (Fig. 2), which has been associated with treatment with platinum-based drugs, which damage DNA by causing covalent adducts[16]. The phased dnSNVs in GEL_9 and GEL_11 are paternally biased (46 paternal:2 maternal, $P = 0.0014$; 28 paternal:1 maternal, $P = 0.012$; binomial test; Extended Data Table 1), and the dnSNVs in GEL_11, who has the largest contribution of SBS31, exhibit a significant transcriptional strand bias, as expected for this signature ($P = 6.9 \times 10^{-6}$, two-sided Poisson test; Extended Data Table 1 and Extended Data Fig. 6). All three fathers had a cancer diagnosis and chemotherapy treatment before conception of their child with a hypermutated genome. The father of GEL_11 was diagnosed with and received chemotherapeutic treatment for osteosarcoma, lung cancer and cancer of the intestinal tract before conception. Cisplatin is a commonly used chemotherapeutic agent for osteosarcoma and lung cancer. Cisplatin mainly reacts with purine bases, forming intrastrand cross-links that can be repaired by NER or bypassed by translesion synthesis, which may in turn induce single-base substitutions[50]. The fathers of GEL_8 and GEL_9 both have a history of testicular cancer where cisplatin is the most commonly administered chemotherapeutic.

GEL_2 and DDD_1 have a similar number of dnSNVs, which are significantly paternally biased (Extended Data Table 1), and share a mutational signature (SBSHYP) that is characterized by an enrichment of C>G and T>G mutations (Fig. 2 and Extended Data Fig. 5) and does not map on to any previously described signatures observed in COSMIC or in response to mutagenic exposure[24,40,51,52] (Supplementary Fig. 3a). The fathers do not share rare non-synonymous variants in any genes. Both fathers received chemotherapy treatment before conception, including nitrogen mustard alkylating agents (Supplementary Table 5), although with different members of this class of chemotherapies. We therefore strongly suspect that this class of chemotherapeutic agents is the cause of this mutational signature. Experimental studies of a subset of alkylating agents have shown them to have diverse mutational signatures[24,51–53] (Supplementary Fig. 3b).

GEL_5 has 182 dnSNVs and a significant paternal bias in the phased dnSNVs ($P = 5.8 \times 10^{-4}$, binomial test; Extended Data Table 1). The father of GEL_5 was diagnosed with systemic lupus erythematosus and received chemotherapy treatment before conception; however, the dnSNVs do not map onto any known chemotherapeutic mutational signatures (Figs. 1b and 2). GEL_5 has a contribution of SBS24, which is associated[4] with aflatoxin exposure in cancer blood samples[22]; however,

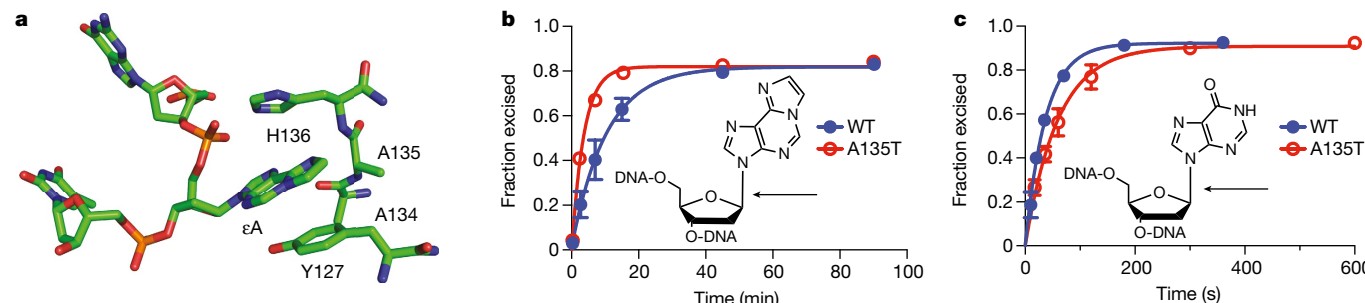

**Fig. 3 | A135T substitution alters the DNA glycosylase activity of MPG.**
**a**, Active-site view of MPG bound to εA-DNA from Protein Data Bank 1EWN. Ala135 and His136 form the binding pocket for the flipped-out base lesion, which is bracketed by Tyr127 on the opposing face. **b**, Single-turnover excision of εA from εA•T is twofold faster for A135T (red) than for wild-type (blue) MPG. **c**, Single-turnover excision of Hx from Hx•T is slower for A135T (red) compared with wild-type (blue) MPG. The arrows indicate the *N*-glycosidic bond that is cleaved by MPG. Data are mean ± s.d. for glycosylase reactions with 10 nM DNA substrate and either 100 nM enzyme for εA excision (*n* = 6) or 40 nM enzyme for Hx excision (*n* = 3) (Extended Data Fig. 9).

aflatoxin exposure is often dietary and there is no evidence of exposure in the father's hospital records

We assessed how parental cancer and exposure to chemotherapy might impact the germline-mutation rate more generally by examining 100kGP hospital records for ICD10 codes related to cancer and chemotherapy recorded before the conception of the child. We identified 27 fathers (0.9%) who had a history of cancer, 7 of whom had testicular cancer (Supplementary Table 7). The offspring of these 27 fathers did not have a significantly increased number of dnSNVs after correcting for parental age (*P* = 0.73, two-sided Wilcoxon test) and their fathers were not significantly older than average (*P* = 0.77, two-sided Wilcoxon test). The available health records are not definitive with regard to historical chemotherapeutic treatments or the potential use of sperm stored before treatment for conception (only 6 had chemotherapy-related ICD10 codes). Although the total number of dnSNVs is not significantly increased, 2 out of the 27 fathers had children with a hypermutated genome, a significant enrichment compared with fathers without a history of cancer (2 out of 27 versus 9 out of 2,891, *P* = 0.0043, Fisher exact test). This is probably a conservative assessment as two other individuals with hypermutation have fathers who were subsequently shown to have had chemotherapy treatment but were not included in this analysis as they did not have any ICD10 codes recorded before conception (Methods). We performed the same analysis across 5,508 mothers in the 100kGP cohort with hospital records before conception and identified 27 mothers (0.5%) with a history of cancer, 9 of whom had chemotherapy recorded. Children of these 27 mothers had a nominally significant increase in dnSNVs after correcting for parental age and data quality (*P* = 0.03, two-sided Wilcoxon test). These mothers were significantly older at the birth of the child compared with the mothers without cancer (*P* = 0.003, Wilcoxon test). Matching on parental age, children of mothers with cancer had a median increase of 9 dnSNVs.

Among the offspring who did not have a hypermutated genome but had a parental history of cancer, we found only one with unusual mutational signatures (Supplementary Fig. 4). PatCancer_10 has 94 dnSNVs (*P* = 0.005, dnSNV *P* value after correcting for parental age) of which 89% phased paternally (Supplementary Fig. 4 and Supplementary Table 7) with a contribution from SBS31, which is associated with platinum-based drugs (Supplementary Fig. 4). Their father was treated for testicular cancer before conception.

## Post-zygotic hypermutation

Two individuals with hypermutation, GEL_4 and GEL_7, had around a fourfold and twofold increase in dnSNVs, respectively, that phase equally between maternal and paternal chromosomes. The VAF of dnSNVs in these individuals was shifted below 0.5 (Extended Data Fig. 7): the

proportion of dnSNVs with VAF < 0.4 was significantly higher compared with all dnSNVs observed (GEL_4: *P* = 3.9 × 10^−59; GEL_7: *P* = 8.3 × 10^−4, two-sided binomial test). These mutations most likely occurred post-zygotically and are not due to a parental hypermutator. Both individuals share a large contribution from SBS1 (ref. [40]) (Fig. 2). GEL_4 has several blood-related clinical phenotypes, including myelodysplasia. The observations in GEL_4 are probably due to clonal haematopoiesis leading to a large number of somatic mutations in the child's blood. We identified a mosaic de novo missense mutation in *ETV6*, a gene that is associated with leukaemia and thrombocytopaenia[54]. We did not observe similar blood-related phenotypes in GEL_7 (although the child was one year old at recruitment), nor did we identify a possible genetic driver of clonal haematopoiesis. We investigated whether a maternal protein with a mutator variant may be affecting the mutation rate in the first few cell divisions. We identified a mosaic maternal missense variant in *TP53* that was previously annotated as pathogenic for Li–Fraumeni cancer predisposition syndrome, which was not observed in the child. It is not known whether this variant is present in the maternal germline or whether it would have a germline mutagenic effect[55].

## Variation in the germline-mutation rate

We investigated the factors influencing the number of dnSNVs per individual in a subset of 7,700 100kGP trios filtered more stringently for data quality (Methods). We estimated that parental age accounts for 69.7% and data quality metrics explain 1.3% of the variance. The variance explained by parental age is smaller than a previous estimate of 95% on the basis of a sample of 78 families[1]. Repeated estimates of the variance explained by parental age from resampling of 78 trios from 100kGP showed that these estimates can vary widely (median = 79%, 95% CI = 52–100%); 7% of resamplings have an estimated variance explained of 95% or greater. We estimated that germline hypermutation in the 11 100kGP individuals with hypermutation explained an additional 7.1% of variance in this cohort. This leaves 21.9% (19.7–23.8%, bootstrap 95% CI) of variance in the numbers of dnSNVs per individual unaccounted for.

Both mutagenic exposures and genetic variation in DNA-repair genes could have a more subtle role in influencing variation in the germline-mutation rate. Moreover, polygenic effects and gene by environment interactions may also contribute. We investigated whether rare variants in DNA-repair genes influence germline-mutation rates in the 100kGP cohort. We curated three sets of rare non-synonymous variants with increasing likelihoods of impacting the germline-mutation rate: (1) variants in all DNA-repair genes (*n* = 186), (2) variants in DNA-repair genes that are most likely to create SNVs (*n* = 66) and (3) the subset of (2) that has been associated with cancer (Methods). We focused on heterozygous variants (MAF < 0.001), but also considered rare homozygous variants (MAF < 0.01) in all DNA-repair genes. There

was no statistically significant effect in any of these groups of variants after Bonferroni correction (Supplementary Fig. 5 and Supplementary Table 8). We examined heterozygous protein-truncating variants (PTVs) in the known cancer mutator gene *MBD4* that are associated with a threefold increased CpG>TpG mutation rate in tumours. We performed whole-genome sequencing of 13 DDD trios with paternal carriers of *MBD4* PTVs. We found no significant increase in either the total number of DNMs or the number of CpG>TpG mutations ($P = 0.56$, $\chi^2$; Supplementary Fig. 6). Power modelling suggested that there is probably not an increase in the CpG germline-mutation rate of higher than a 22%.

To examine potential polygenic contributions, we estimated the residual variation in the number of dnSNVs in the 100kGP cohort (after correcting for parental age, data quality and hypermutation status) explained by more common genetic variants. We estimated this separately for fathers and mothers using GREML-LDMS[56] stratified by minor allele frequency and linkage disequilibrium. We found that maternal germline variation (MAF > 0.001) is unlikely to explain much residual variation ($h^2 = 0.07$, $P = 0.21$, GCTA reported results; Supplementary Table 9). We found that paternal variation could contribute a substantial fraction of residual variation ($h^2 = 0.53$, 95% CI = 0.20–0.85, $P = 0.09$); however, this seems to be concentrated exclusively in low-frequency variants (0.001 < MAF < 0.01, $h^2 = 0.52$, 95% CI = 0.01–0.94) rather than more common variants (MAF > 0.01, $h^2 = 0.008$, 95% CI = 0–0.38; Supplementary Table 9). Further investigation of polygenic contributions will require larger sample sizes.

## Discussion

Germline hypermutation is an uncommon but important phenomenon. We identified 12 individuals with hypermutation from over 20,000 parent–offspring sequenced trios in the DDD and 100kGP cohorts with a two- to sevenfold increase in the number of dnSNVs. There are probably other individuals with germline hypermutation in the DDD cohort, as screening this exome-sequenced cohort for potential individuals with hypermutation for confirmation by genome sequencing will have missed some individuals with two- to sevenfold hypermutation.

In two individuals with hypermutation, the excess mutations occurred post-zygotically; however, for the majority ($n = 8$), excess dnSNVs phased paternally, implicating the father as the source of hypermutation. For five of these fathers, mutational signatures and clinical records implicated the mutagenicity of two classes of chemotherapeutics: platinum-based drugs ($n = 3$) and mustard-derived alkylating agents ($n = 2$). For two fathers, functional and clinical data implicated the mutagenicity of homozygous missense variants in the known DNA-repair genes *XPC* and *MPG*.

Our findings imply that defects in DNA-repair genes can increase germline-mutation rates in addition to their well-established impacts on somatic mutation rates[57]. However, DNA-repair defects do not always behave similarly in the soma and the germline. We found that PTVs in an established somatic mutator gene, *MBD4*, did not have a detectable effect in the germline[58]. We also did not observe a significant effect on germline-mutation rates of rare non-synonymous variants in DNA-repair genes more generally. Paternal variants previously associated with cancer had a nominally significant effect but amounted to an average increase of only around 2 dnSNVs. Both larger sample sizes and additional variant curation will probably be needed to investigate this further. Genes and pathways that impact germline mutation more than the soma may also exist; detecting mutagenic variants in these genes will be challenging.

Germline hypermutation accounted for 7% of the variance in the germline-mutation rate in the 100kGP cohort. The ascertainment in this cohort for rare genetic diseases probably means that individuals with germline hypermutation are enriched relative to the general population. As a consequence, our estimate of the contribution of germline

hypermutation is probably inflated. However, the absolute risk of an individual with a hypermutated germline having a child with a genetic disease is low. The population average risk for having a child with a severe developmental disorder caused by a DNM has been estimated to be 1 in 300 births[12] and so a fourfold increase in DNMs in a child would increase this absolute risk to just over 1%. Thus, most individuals with germline hypermutation will not have a genetic disease, and germline hypermutation should also be observed in healthy individuals.

The two genetic causes of germline hypermutation that we identified were both recessive in action. Similarly, most DNA-repair disorders act recessively in their cellular mutagenic effects. This implies that genetic causes of germline hypermutation are likely to arise at substantially higher frequencies in populations with high rates of parental consanguinity. In such populations, the overall incidence of germline hypermutation may be higher, and the proportion of variance in the number of dnSNVs per offspring accounting for genetic effects will be higher. We anticipate that studies focused on these populations are likely to identify additional mutations that affect germline-mutation rate.

We found that, among 7,700 100kGP families, parental age explained only around 70% of the variance in the numbers of dnSNVs per offspring, which is substantially smaller than a previous estimate of 95% based on 78 families[1]. Resampling analyses showed that, in small numbers of families, estimates of the variance explained by parental age have wide confidence intervals such that these two estimates are not inconsistent, although estimates based on a two order of magnitude greater number of samples will be much more precise. A residual ~20% of variation in the numbers of germline dnSNVs per individual remains unexplained by parental age, data quality and hypermutation. We found that neither rare variants in known DNA-repair genes nor polygenic contributions from common variants (MAF > 0.01) are likely to account for a large proportion of this unexplained variance. Larger sample sizes are required to further evaluate polygenic contributions from intermediate frequency (0.001 < MAF < 0.01) variants. A limitation of these heritability analyses is the use of DNMs in offspring as a proxy for germline-mutation rates in individual parents. Measuring germline-mutation rates more directly by, for example, sequencing hundreds of single gametes per individual, should facilitate better powered association studies and heritability analyses.

Environmental exposures are also likely to contribute to germline-mutation rate variation. We have observed evidence that certain chemotherapeutic agents can affect the germline-mutation rate. Targeted studies on the germline mutagenic effects of different chemotherapeutic agents (such as in cancer survivor cohorts) will be crucial in understanding this further. We anticipate heterogeneity in the germline mutagenic effects of different chemotherapeutic agents, in part due to differences in the permeability of the blood–testis barrier[59], as well as variation in the vulnerability to chemotherapeutic germline mutagenesis by sex and age. As few individuals receive chemotherapy before reproduction, chemotherapeutic exposures will not explain a large proportion of the remaining variation in germline-mutation rates. However chemotherapeutic mutagenesis has important implications for patients receiving some chemotherapies who plan to have children, especially in relation to storing unexposed gametes for future use of assisted reproductive technologies.

Unexplained hypermutation and additional variance in the germline-mutation rate might be explained by other environmental exposures. One limitation of this study was the lack of data on non-therapeutic environmental exposures. Reassuringly, the narrow distribution of DNMs per individual in the 100kGP cohort suggests that it is unlikely that there are common environmental mutagen exposures in the UK (such as cigarette smoking) that cause a substantive (for example, >1.5 times) fold increase in mutation rates and concomitant disease risk. The germline generally appears to be well protected from large increases in mutation rate. However, including a broader spectrum

of environmental exposures in future studies would help to identify more subtle effects and may reveal gene-by-environment interactions.

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

**Genomics England Research Consortium**

**Loukas Moutsianas⁵ & Chris Odhams⁵**

⁵Genomics England, London, UK.

# Methods

## DNM filtering in 100,000 Genomes Project

We analysed DNMs called in 13,949 parent–offspring trios from 12,609 families from the rare disease programme of the 100,000 Genomes Project. The rare disease cohort includes individuals with a wide array of diseases, including neurodevelopmental disorders, cardiovascular disorders, renal and urinary tract disorders, ophthalmological disorders, tumour syndromes, ciliopathies and others. These are described in more detail in previous publications[60,61]. The cohort was whole-genome sequenced at around 35× coverage and variant calling for these families was performed through the Genomics England rare disease analysis pipeline. The details of sequencing and variant calling have been previously described[61]. DNMs were called by the Genomics England Bioinformatics team using the Platypus variant caller[62]. These were selected to optimize various properties, including the number of DNMs per person being approximately what we would expect, the distribution of the VAF of the DNMs to be centred around 0.5 and the true positive rate of DNMs to be sufficiently high as calculated from examining IGV plots. The filters applied were as follows:

- Genotype is heterozygous in child (1/0) and homozygous in both parents (0/0).
- Child read depth (RD) > 20, mother RD > 20, father RD > 20.
- Remove variants with >1 alternative read in either parent.
- VAF > 0.3 and VAF < 0.7 for child.
- Remove SNVs within 20 bp of each other. Although this is probably removing true MNVs, the error mode was very high for clustered mutations.
- Removed DNMs if child RD > 98 (ref. [14]).
- Removed DNMs that fell within known segmental duplication regions as defined by the UCSC (http://humanparalogy.gs.washington.edu/build37/data/GRCh37GenomicSuperDup.tab).
- Removed DNMs that fell in highly repetitive regions (http://human-paralogy.gs.washington.edu/build37/data/GRCh37simpleRepeat.txt).
- For DNM calls that fell on the X chromosome, these slightly modified filters were used:
  - For DNMs that fell in PAR regions, the filters were unchanged from the autosomal calls apart from allowing for both heterozygous (1/0) and hemizygous (1) calls in males.
  - For DNMs that fell in non-PAR regions the following filters were used:
    - For males: RD > 20 in child, RD > 20 in mother, no RD filter on father.
    - For males: the genotype must be hemizygous (1) in child and homozygous in mother (0/0).
    - For females: RD > 20 in child, RD > 20 in mother, RD > 10 in father.

## DNM filtering in DDD

To identify individuals with hypermutation in the DDD study, we started with exome-sequencing data from the DDD study of families with a child with a severe, undiagnosed developmental disorder. The recruitment of these families has been described previously[63]: families were recruited at 24 clinical genetics centres within the UK National Health Service and the Republic of Ireland. Families gave informed consent to participate, and the study was approved by the UK Research Ethics Committee (10/H0305/83, granted by the Cambridge South Research Ethics Committee, and GEN/284/12, granted by the Republic of Ireland Research Ethics Committee). Sequence alignment and variant calling of SNVs and indels were conducted as previously described. DNMs were called using DeNovoGear and filtered as described previously[12,64]. The analysis in this paper was conducted on a subset (7,930 parent–offspring trios) of the full current cohort, which was not available at the start of this research.

In the DDD study, we identified 9 individuals out of 7,930 parent–offspring trios with an increased number of exome DNMs after accounting for parental age (7-17 exome DNMs compared to an expected number of ~2). These were subsequently submitted along with their parents for PCR-free whole-genome sequencing at >30x mean coverage using Illumina 150bp paired end reads and in house WSI sequencing pipelines. Reads were mapped with bwa (v0.7.15)[65]. DNMs were called from these trios using DeNovoGear[64] and were filtered as follows:

- Child RD > 10, mother RD > 10, father RD > 10.
- Alternative allele RD in child of >2.
- Filtered on strand bias across parents and child (p-value > 0.001, Fisher's exact test).
- Removed DNMs that fell within known segmental duplication regions as defined by the UCSC (http://humanparalogy.gs.washington.edu/build37/data/GRCh37GenomicSuperDup.tab).
- Removed DNMs that fell in highly repetitive regions (http://human-paralogy.gs.washington.edu/build37/data/GRCh37simpleRepeat.txt).
- Allele frequency in gnomAD < 0.01.
- VAF < 0.1 for both parents.
- Removed mutations if both parents have >1 read supporting the alternative allele.
- Test to see whether VAF in the child is significantly greater than the error rate at that site as defined by error sites estimated using Shearwater[66].
- Posterior probability from DeNovoGear > 0.00781 (refs. [12,64]).
- Removed DNMs if the child RD > 200.

After applying these filters, this resulted in 1,367 DNMs. All of these DNMs were inspected in the Integrative Genome Viewer[67] and removed if they appeared to be false-positives. This resulted in a final set of 916 DNMs across the 9 trios. One out of the nine had 277 dnSNVs genome wide, whereas the others had expected numbers (median, 81 dnSNVs).

## Parental phasing of DNMs

To phase the DNMs in both 100kGP and DDD, we used a custom script that used the following read-based approach to phase a DNM. This first searches for heterozygous variants within 500 bp of the DNM that was able to be phased to a parent (so not heterozygous in both parents and offspring). We next examined the reads or read pairs that included both the variant and the DNM and counted how many times we observed the DNM on the same haplotype of each parent. If the DNM appeared exclusively on the same haplotype as a single parent then that was determined to originate from that parent. We discarded DNMs that had conflicting evidence from both parents. This code is available on GitHub (https://github.com/queenjobo/PhaseMyDeNovo).

## Parental age and germline-mutation rate

To assess the effect of parental age on germline-mutation rate, we ran the following regressions on autosomal DNMs. These and subsequent statistical analyses were performed primarily in R (v.4.0.1). On all (unphased) DNMs, we ran two separate regressions for SNVs and indels. We chose a negative binomial generalized linear model (GLM) here as the Poisson was found to be overdispersed. We fitted the following model using a negative Binomial GLM with an identity link where $Y$ is the number of DNMs for an individual:

$$E(Y) = \beta_0 + \beta_1\text{paternal age} + \beta_2\text{maternal age}$$

For the phased DNMs we fit the following two models using a negative binomial GLM with an identity link where $Y_{\text{maternal}}$ is the number of maternally derived DNMs and $Y_{\text{paternal}}$ is the number of paternally derived DNMs:

$$E(Y_{\text{paternal}}) = \beta_0 + \beta_1\text{paternal age}$$
$$E(Y_{\text{maternal}}) = \beta_0 + \beta_1\text{maternal age}$$

## Individuals with hypermutation in the 100kGP cohort

To identify individuals with hypermutation in the 100kGP cohort, we first wanted to regress out the effect of parental age as described in the parental age analysis. We then looked at the distribution of the studentized residuals and then, assuming these followed a $t$ distribution with $N-3$ degrees of freedom, calculated a $t$-test P value for each individual. We took the same approach for the number of indels except, in this case, $Y$ would be the number of de novo indels.

We identified 21 individuals out of 12,471 parent–offspring trios with a significantly increased number of dnSNVs genome wide ($P < 0.05/12,471$ tests). We performed multiple quality control analyses, which included examining the mutations in the Integrative Genomics Browser for these individuals to examine DNM calling accuracy, looking at the relative position of the DNMs across the genome and examining the mutational spectra of the DNMs to identify any well-known sequencing error mutation types. We identified 12 that were not truly hypermutated. The majority of false-positives (10) were due to a parental somatic deletion in the blood, increasing the number of apparent DNMs (Supplementary Fig. 7). These individuals had some of the highest numbers of DNMs called (up to 1,379 DNMs per individual). For each of these 10 individuals, the DNM calls all clustered to a specific region in a single chromosome. In this same corresponding region in the parent, we observed a loss of heterozygosity when calculating the heterozygous/homozygous ratio. Moreover, many of these calls appeared to be low-level mosaic in that same parent. This type of event has previously been shown to create artifacts in CNV calls and is referred to as a 'loss of transmitted allele' event[68]. The remaining two false-positives were due to bad data quality in either the offspring or one of the parents leading to poor DNM calls. The large number of DNMs in these false-positive individuals also led to significant underdispersion in the model so, after removing these 12 individuals, we reran the regression model and subsequently identified 11 individuals who appeared to have true hypermutation ($P < 0.05/12,459$ tests).

## Extraction of mutational signatures

Mutational signatures were extracted from maternally and paternally phased autosomal DNMs, 24 controls (randomly selected), 25 individuals (father with a cancer diagnosis before conception), 27 individuals (mother with a cancer diagnosis before conception) and 12 individuals with hypermutation that we identified. All DNMs were lifted over to GRCh37 before signature extraction (100kGP samples are a mix of GRCh37 and GRCh38) and, through the liftover process, a small number of 100kGP DNMs were lost (0.09% overall, 2 DNMs were lost across all of the individuals with hypermutation). The mutation counts for all of the samples are shown in Supplementary Table 1. This was performed using SigProfiler (v.1.0.17) and these signatures were extracted and subsequently mapped on to COSMIC mutational signatures (COSMIC v.91, Mutational Signature v.3.1)[19,40]. SigProfiler defaults to selecting a solution with higher specificity than sensitivity. A solution with 4 de novo signatures was chosen as optimal by SigProfiler for the 12 individuals with germline-hypermutated genomes. Another stable solution with five de novo signatures was also manually deconvoluted, which has been considered as the final solution. The mutation probability for mutational signature SBSHYP is shown in Supplementary Table 3.

## External exposure signature comparison

We compared the extracted signatures from these individuals with hypermutation with a compilation of previously identified signatures caused by environmental mutagens from the literature. The environmental signatures were compiled from refs. [24,51,52]. Comparison was calculated as the cosine similarity between the different signatures.

## Genes involved in DNA repair

We compiled a list of DNA-repair genes that were taken from an updated version of the table in ref. [69] (https://www.mdanderson.org/documents/Labs/Wood-Laboratory/human-dna-repair-genes.html). These can be found in Supplementary Table 4. These are annotated with the pathways that they are involved with (such as nucleotide-excision repair, mismatch repair). A 'rare' variant is defined as those with an allele frequency of <0.001 for heterozygous variants and those with an allele frequency of <0.01 for homozygous variants in both the 1000 Genomes as well as across the 100kGP cohort.

## Kinetic characterization of MPG

The A135T variant of *MPG* was generated by site-directed mutagenesis and confirmed by sequencing both strands. The catalytic domain of WT and A135T *MPG* was expressed in BL21(DE3) Rosetta2 *Escherichia coli* and purified as described for the full-length protein[70]. Protein concentration was determined by absorbance at 280 nm. Active concentration was determined by electrophoretic mobility shift assay with 5′-FAM-labelled pyrolidine-DNA[48] (Extended Data Fig. 8). Glycosylase assays were performed with 50 mM NaMOPS, pH 7.3, 172 mM potassium acetate, 1 mM DTT, 1 mM EDTA, 0.1 mg ml$^{-1}$ BSA at 37 °C. For single-turnover glycosylase activity, a 5'-FAM-labelled duplex was annealed by heating to 95 °C and slowly cooling to 4 °C (Extended Data Fig. 9). DNA substrate concentration was varied between 10 nM and 50 nM, and MPG concentration was maintained in at least twofold excess over DNA from 25 nM to 10,000 nM. Samples taken at timepoints were quenched in 0.2 M NaOH, heated to 70 °C for 12.5 min, then mixed with formamide/EDTA loading buffer and analysed by 15% denaturing polyacrylamide gel electrophoresis. Fluorescence was quantified using the Typhoon 5 imager and ImageQuant software (GE). The fraction of product was fit by a single exponential equation to determine the observed single-turnover rate constant ($k_{obs}$). For Hx excision, the concentration dependence was fit by the equation $k_{obs} = k_{max} [E]/(K_{1/2} + [E])$, where $K_{1/2}$ is the concentration at which half the maximal rate constant ($k_{max}$) was obtained and [E] is the concentration of enzyme. It was not possible to measure the $K_{1/2}$ for εA excision using a fluorescence-based assay owing to extremely tight binding[71]. Multiple turnover glycosylase assays were performed with 5 nM MPG and 10–40-fold excess of substrate (Extended Data Fig. 8).

## Fraction of variance explained

To estimate the fraction of germline mutation variance explained by several factors, we fit the following negative binomial GLMs with an identity link. Data quality is likely to correlate with the number of DNMs detected so, to reduce this variation, we used a subset of the 100kGP dataset that had been filtered on some base quality control metrics by the Bioinformatics team at GEL:
- Cross-contamination < 5%
- Mapping rate > 75%
- Mean sample coverage > 20
- Insert size < 250

We then included the following variables to try to capture as much of the residual measurement error which may also be impacting DNM calling. In brackets are the corresponding variable names used in the models below:
- Mean coverage for the child, mother and father (child mean RD, mother mean RD, father mean RD)
- Proportion of aligned reads for the child, mother and father (child prop aligned, mother prop aligned, father prop aligned)
- Number of SNVs called for child, mother and father (child snvs, mother snvs, father snvs)
- Median VAF of DNMs called in child (median VAF)
- Median 'Bayes Factor' as outputted by Platypus for DNMs called in the child. This is a metric of DNM quality (median BF).

The first model only included parental age:

$E(Y) = \beta_0 + \beta_1\text{paternal age} + \beta_2\text{maternal age}$

The second model also included data quality variables as described above:

$$
\begin{aligned}
E(Y) =\ & \beta_0 + \beta_1\text{paternal age} + \beta_2\text{maternal age} \\
& + \beta_3\text{child mean RD} + \beta_4\text{mother mean RD} \\
& + \beta_5\text{father mean RD} + \beta_6\text{child prop aligned} \\
& + \beta_7\text{mother prop aligned} \\
& \quad + \beta_8\text{father prop aligned} \\
& + \beta_9\text{childs nvs} + \beta_{10}\text{mother snvs} \\
& \quad + \beta_{11}\text{father snvs} \\
& + \beta_{12}\text{median VAF} + \beta_{13}\text{median BF}
\end{aligned}
$$

The third model included a variable for excess mutations in the 11 confirmed individuals with hypermutation (hm excess) in the 100kGP dataset. This variable was the total number of mutations subtracted by the median number of DNMs in the cohort (65), $Y_{\text{hypermutated}} - \text{median}(Y)$ for these 11 individuals and 0 for all other individuals.

$$
\begin{aligned}
E(Y) =\ & \beta_0 + \beta_1\text{paternal age} + \beta_2\text{maternal age} \\
& + \beta_3\text{child mean RD} + \beta_4\text{mother mean RD} \\
& + \beta_5\text{father mean RD} + \beta_6\text{child prop aligned} \\
& + \beta_7\text{mother prop aligned} \\
& \quad + \beta_8\text{father prop aligned} \\
& + \beta_9\text{child snvs} + \beta_{10}\text{mother snvs} \\
& \quad + \beta_{11}\text{father snvs} \\
& + \beta_{12}\text{median VAF} + \beta_{13}\text{median BF} \\
& \quad + \beta_{14}\text{hm excess}
\end{aligned}
$$

The fraction of variance ($F$) explained after accounting for Poisson variance in the mutation rate was calculated in a similar way to in ref.[1] using the following formula:

$$ F = \text{pseudo } R^2 \frac{1 - \overline{Y}}{\text{Var}(Y)} $$

McFadden's pseudo $R^2$ was used here as a negative binomial GLM was fitted. We repeated these analyses fitting an ordinary least squares regression, as was done in ref.[1], using the $R^2$ and got comparable results. To calculate a 95% confidence interval, we used a bootstrapping approach. We sampled with a replacement 1,000 times and extracted the 2.5% and 97.5% percentiles.

## Rare variants in DNA-repair genes

We fit eight separate regressions to assess the contribution of rare variants in DNA-repair genes (compiled as described previously). These were across three different sets of genes: variants in all DNA-repair genes, variants in a subset of DNA-repair genes that are known to be associated with base-excision repair, MMR, NER or a DNA polymerase, and variants within this subset that have also been associated with a cancer phenotype. For this, we downloaded all ClinVar entries as of October 2019 and searched for germline 'pathogenic' or 'likely pathogenic' variants annotated with cancer[55]. We tested both all non-synonymous variants and just PTVs for each set. To assess the contribution of each of these sets, we created two binary variables per set indicating a presence or absence of a maternal or paternal variant for each individual, and then ran a negative binomial regression for each subset including these

as independent variables along with hypermutation status, parental age and quality-control metrics as described in the previous section.

## Simulations for parental age effect

We downsampled from the full cohort to examine how the estimates of the fraction of variance in the number of DNMs explained by paternal age varied with sample number. We first simulated a random sample as follows 10,000 times:
- Randomly sample 78 trios (the number of trios in ref.[1].)
- Fit ordinary least squares of $E(Y) = \beta_0 + \beta_1\text{paternal age}$.
- Estimated the fraction of variance ($F$) as described in ref.[1].

We found that the median fraction explained was 0.77, with a s.d. of 0.13 and with 95% of simulations fallings between 0.51 and 1.00.

## Parental cancer diagnosis before conception

To identify parents who had received a cancer diagnosis before the conception of their child, we examined the admitted patient care hospital episode statistics of these parents. There were no hospital episode statistics available before 1997, and many individuals did not have any records until after the birth of the child. To ensure that comparisons were not biased by this, we first subset to parents who had at least one episode statistic recorded at least two years before the child's year of birth. Two years before the child's birth was our best approximation for before conception without the exact child date of birth. This resulted in 2,891 fathers and 5,508 mothers. From this set we then extracted all entries with ICD10 codes with a 'C' prefix, which corresponds to malignant neoplasms, and 'Z85', which corresponds to a personal history of malignant neoplasm. We defined a parent as having a cancer diagnosis before conception if they had any of these codes recorded ≥2 years before the child's year of birth. We also extracted all entries with ICD10 code 'Z511', which codes for an 'encounter for antineoplastic chemotherapy and immunotherapy'.

Two fathers of individuals with hypermutation who we suspect had chemotherapy before conception did not meet these criteria as the father of GEL_5 received chemotherapy for treatment for systemic lupus erythematosus and not cancer and, for the father of GEL_8, the hospital record 'personal history of malignant neoplasm' was entered after the conception of the child (Supplementary Table 5).

To compare the number of dnSNVs between the group of individuals with parents with and without cancer diagnoses, we used a Wilcoxon test on the residuals from the negative binomial regression on dnSNVs correcting for parental age, hypermutation status and data quality. To look at the effect of maternal cancer on dnSNVs, we matched these individuals on maternal and paternal age with sampling replacement with 20 controls for each of the 27 individuals. We found a significant increase in DNMs (74 compared to 65 median dnSNVs, $P = 0.001$, Wilcoxon Test).

## SNP heritability analysis

For this analysis, we started with the same subset of the 100kGP dataset that had been filtered as described in the analysis of the impact of rare variants in DNA-repair genes across the cohort (see above). To ensure variant quality, we subsetted to variants that have been observed in genomes from gnomAD (v.3)[72]. These were then filtered by ancestry to parent–offspring trios where both the parents and child mapped on to the 1000 Genomes GBR subpopulations. The first 10 principal components were subsequently included in the heritability analyses. To remove cryptic relatedness, we removed individuals with an estimated relatedness of >0.025 (using GCTA grm-cutoff, 0.025). This resulted in a set of 6,352 fathers and 6,329 mothers. The phenotype in this analysis was defined as the residual from the negative binomial regression of the number of DNMs after accounting for parental age, hypermutation status and several data quality variables, as described when estimating the fraction of DNM count variation explained (see above). To estimate heritability, we ran GCTA GREML-LDMS on two linkage disequilibrium

stratifications and three MAF bins (0.001–0.01, 0.01–0.05, 0.05–1)[56]. For mothers, this was run with the --reml-no-constrain option because it would otherwise not converge (Supplementary Table 9).

## Reporting summary

Further information on research design is available in the Nature Research Reporting Summary linked to this paper.

## Data availability

Sequence and variant-level data and phenotypic data for the DDD study data are available from the European Genome–Phenome Archive (EGA: EGAS00001000775). The DDD_1 WGS and DNM data are under EGAD00001008497. These data are under managed access to ensure that the work proposed by the researchers is allowed under the study's ethical approval. Sequence- and variant-level data (including the DNM dataset) and phenotypic data from the 100,000 Genomes Project can be accessed by application to Genomics England following the procedure outlined at https://www.genomicsengland.co.uk/about-gecip/joining-research-community/. Other databases are available online: Genome Aggregation Database (gnomAD v.2.1.1; https://gnomad.broadinstitute.org/); Catalogue of Somatic Mutations in Cancer (v.3.1; https://cancer.sanger.ac.uk/); ClinVar (https://www.ncbi.nlm.nih.gov/clinvar/).

## Code availability

Phasing of mutations was performed with a custom Python (3) script available at GitHub (https://github.com/queenjobo/PhaseMyDeNovo).

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

**Acknowledgements** We thank the families and their clinicians for their participation and engagement, and our colleagues who assisted in the generation and processing of data; M. Stratton, P. Campbell, E. Mitchell, E. Dunstone, H. Martin, K. Chundru, M. Przeworski and J. Korbel for discussions and advice. This research was made possible through access to the data and findings generated by the 100,000 Genomes Project. The 100,000 Genomes Project is managed by Genomics England Limited (a wholly owned company of the Department of Health and Social Care). The 100,000 Genomes Project is funded by the National Institute for Health Research and NHS England. The Wellcome Trust, Cancer Research UK and the Medical Research Council have also funded research infrastructure. The 100,000 Genomes Project uses data provided by patients and collected by the National Health Service as part of their care and support. The DDD study presents independent research commissioned by the Health Innovation Challenge Fund (grant no. HICF-1009-003). The full acknowledgements can be found online (www.ddduk.org/access.html). This research was funded in part by the Wellcome Trust (grant no. 206194). For the purpose of open access, the authors have applied a CC-BY public copyright licence to any author accepted manuscript version arising from this submission. This work was supported by Health Data Research UK, which is funded by the UK Medical Research Council, Engineering and Physical Sciences Research Council, Economic and Social Research Council, Department of Health and Social Care (England), Chief Scientist Office of the Scottish Government Health and Social Care Directorates, Health and Social Care Research and Development Division (Welsh Government), Public Health Agency (Northern Ireland), British Heart Foundation and Wellcome.

**Author contributions** J.K. and M.H. conceived the project. J.K., C.O., L.M., P.D., E.P., J.M., G.G. and P.S. contributed to the generation and quality control of data. J.K., R.S., M.N., T.C., B.I. and P.O. performed analyses/experiments and contributed to the generation of figures. J.C., A.B. and H.F. provided clinical data and interpretation. M.H., R.R and P.O. provided experimental and analytical supervision. J.K., R.R. and M.H. wrote the manuscript with input from all of the authors. M.H. supervised the project.

**Competing interests** M.H. is a co-founder of, consultant to and holds shares in Congenica, a genetics diagnostic company. L.M. and C.O. are employees of Genomics England Ltd.

**Additional information**
**Correspondence and requests for materials** should be addressed to Matthew Hurles.

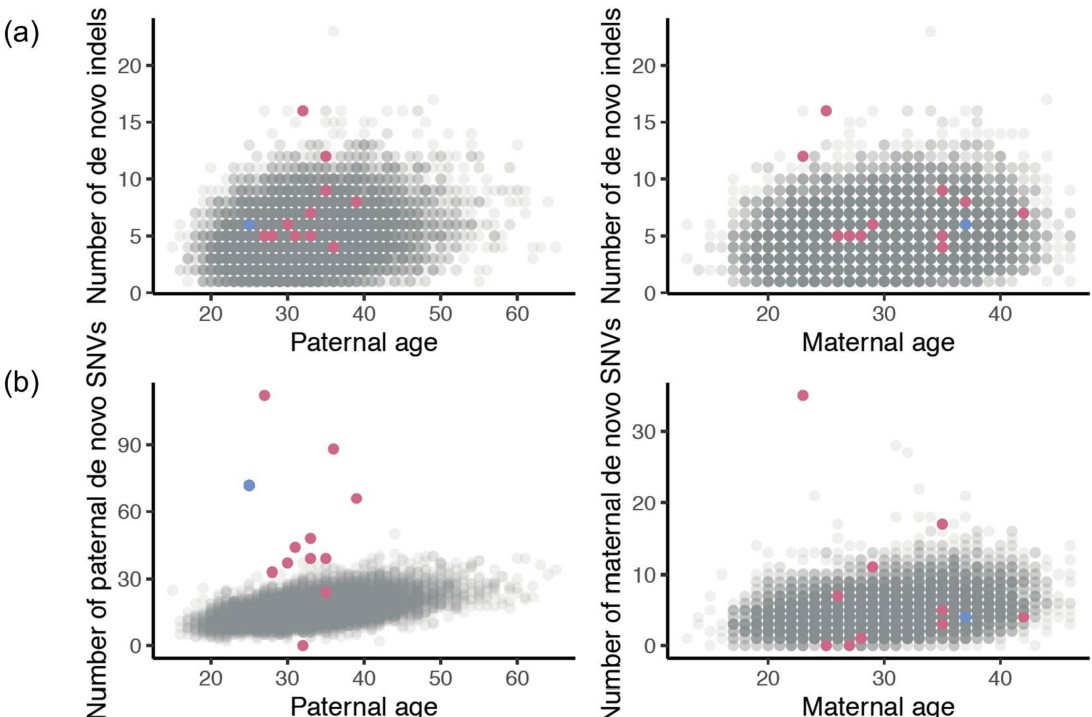

**Extended Data Fig. 1 | Parental age and number of DNMs. (a)** Paternal and maternal age against the number of dnInDels. **(b)** Paternal age against number of paternally phased dnSNVs and maternal age against number of maternally phased dnSNVs. Hypermutated individuals are highlighted in pink (11 individuals in 100kGP) and blue (DDD individuals).

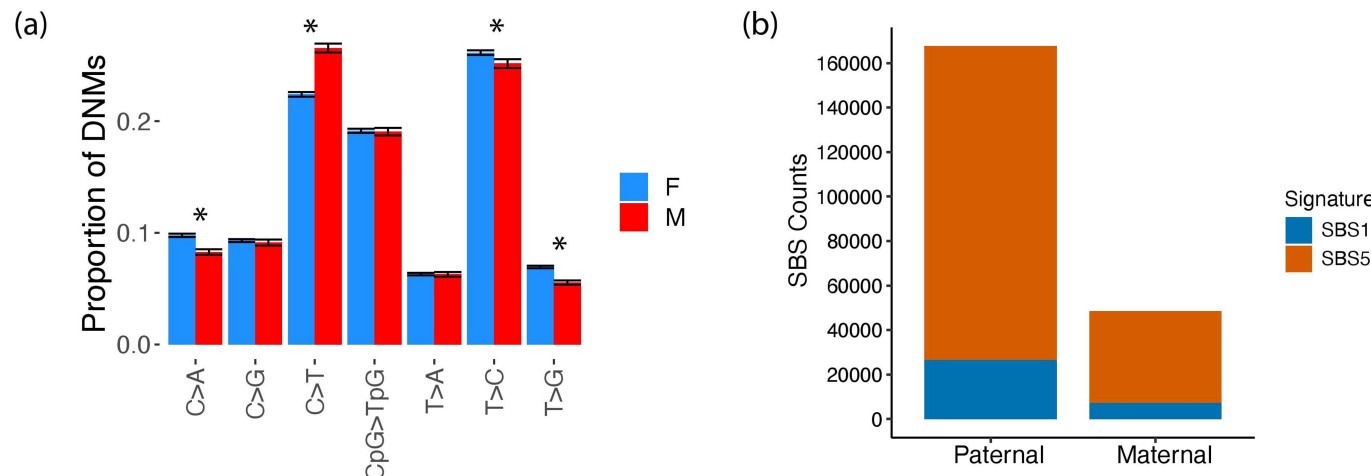

(a)

(b)

**Extended Data Fig. 2 | Mutational spectra and signatures for maternal vs paternal DNMs across 100kGP cohort. (a)** Mutational spectra for maternal vs paternal DNMs across 100kGP cohort (48,381 maternal DNMs and 167,558 paternal DNMs). Significant differences (chi-squared test, two sided, Bonferroni corrected threshold of *P* < 0.05/7) are marked with * (p-values:

C > A 4.6310-23, C > G 0.20, C > T 3.2510-80, CpG>TpG 0.75, T > A 0.98, T > C1.6210-5, T > G 6.8110-28). The 95% confidence intervals are shown. **(b)** Mutational signature decomposition for DNMs in maternally and paternally derived DNMs. Signatures extracted with SigProfiler. Colours correspond to COSMIC signatures.

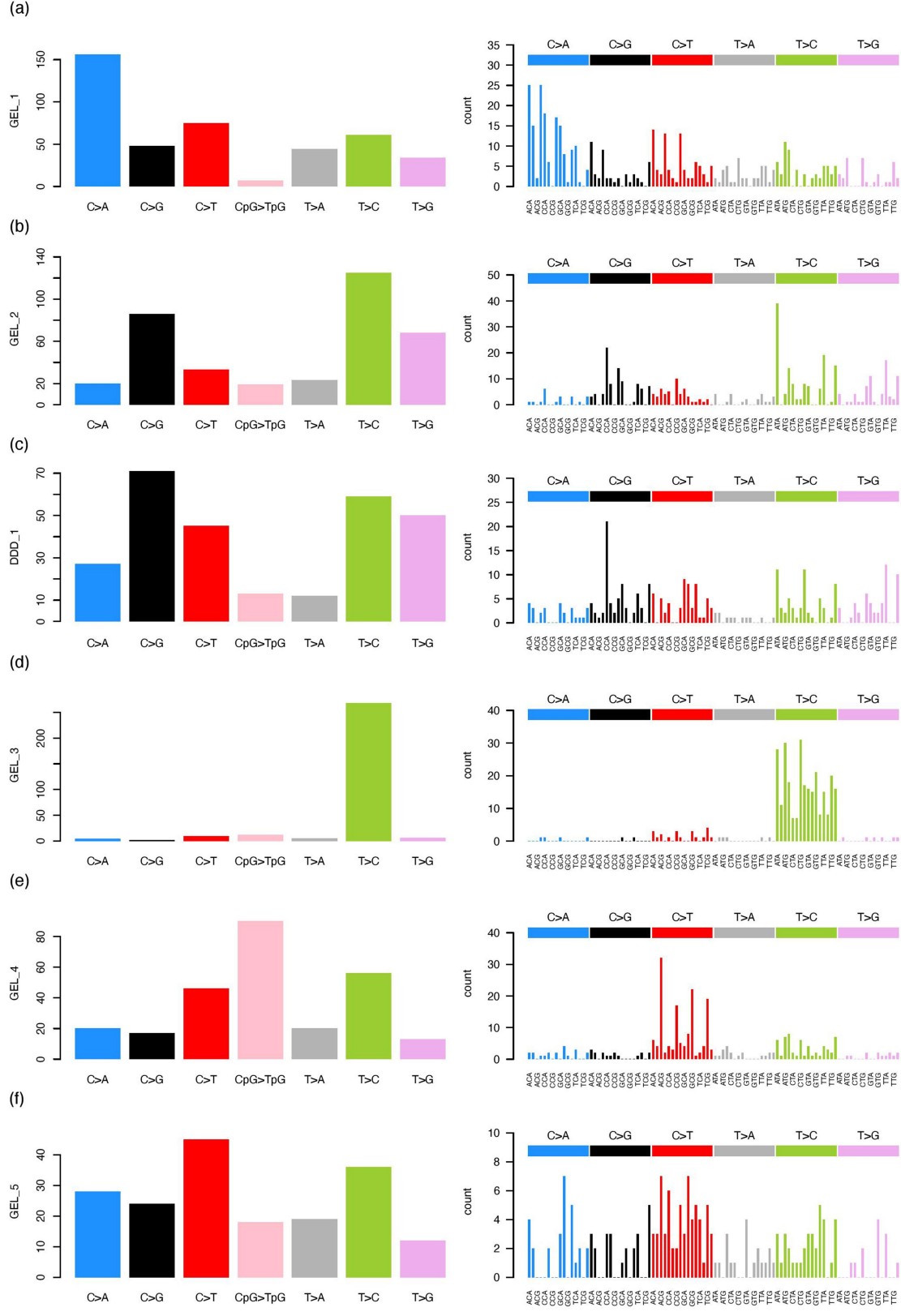

**Extended Data Fig. 3 | Mutational spectra for the DNMs of hypermutated individuals part 1.** (**a**–**f** correspond to individual GEL_1, GEL_2, DDD_1, GEL_3, GEL_4, and GEL_5 respectively). Each row is a hypermutated individual showing the mutational spectra according to count of mutations per each single base change (with CpG>TpG mutations separated from other C>T mutations) and the second plot is the mutation count for all 96 mutations in their trinucleotide context. The x-axis demonstrates the reference trinucleotide sequence with the mutated base highlighted. The colour and label on the bar above indicates the mutation type.

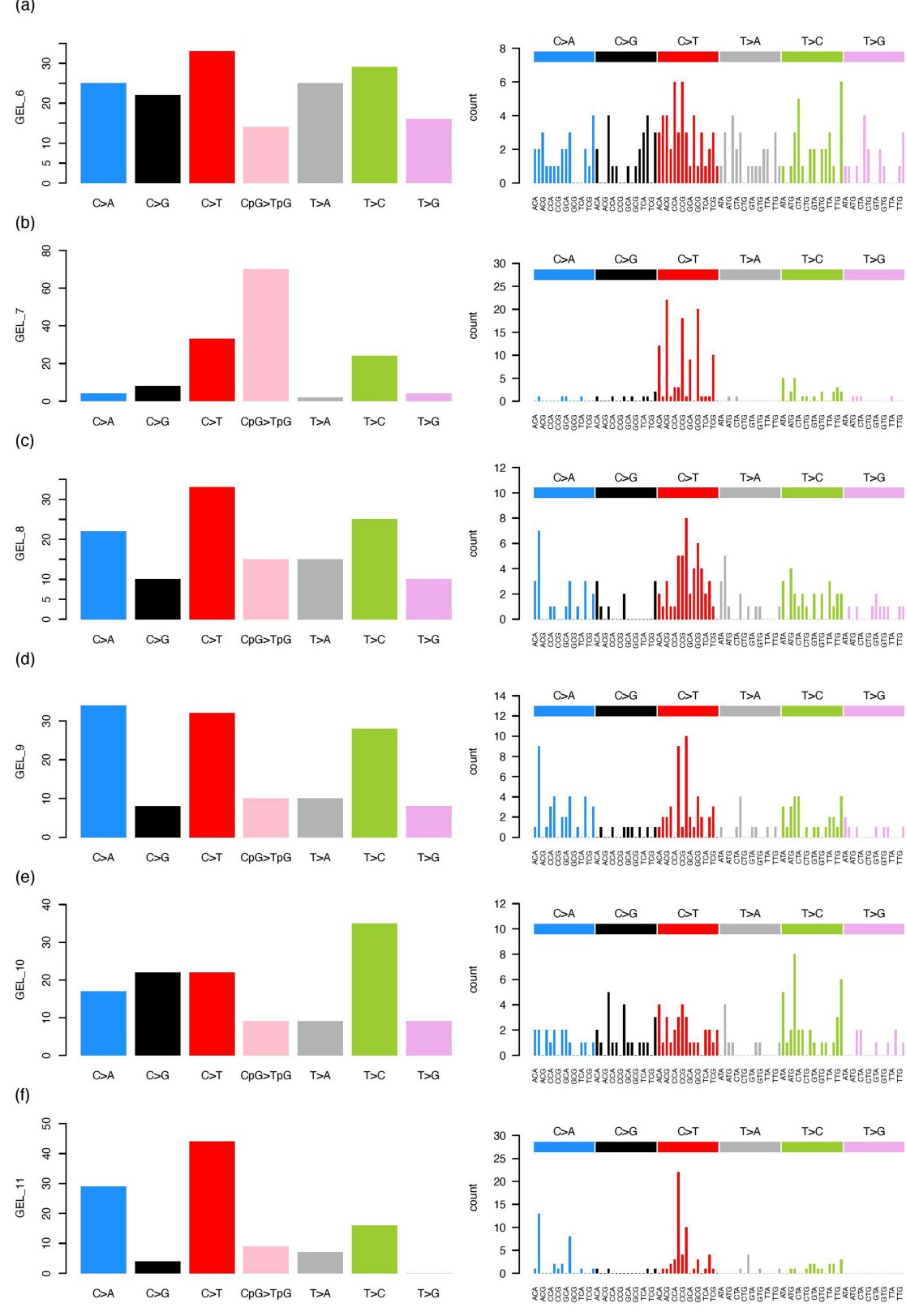

**Extended Data Fig. 4 | Mutational spectra for the DNMs hypermutated individuals part 2.** (a–f correspond to individual GEL_6, GEL_7, GEL_8, GEL_9, GEL_10 and GEL_11 respectively). Each row is a hypermutated individual showing the mutational spectra according to count of mutations per each single base change (with CpG>TpG mutations separated from other C>T mutations) and the second plot is the mutation count for all 96 mutations in their trinucleotide context. The x-axis demonstrates the reference trinucleotide sequence with the mutated base highlighted. The colour and label on the bar above indicates the mutation type.

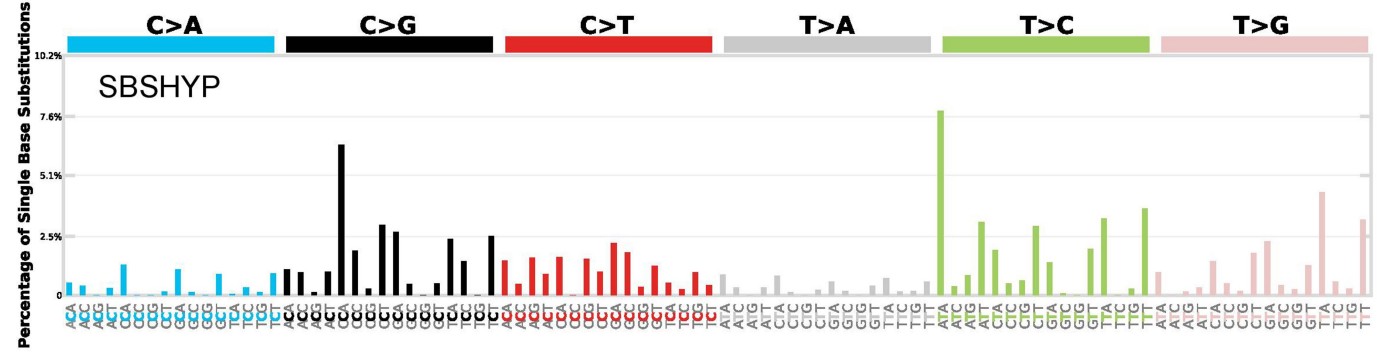

**Extended Data Fig. 5 | Novel mutational signature SBSHYP.** Trinucleotide context mutational profile of novel extracted mutational signature SBSHYP.

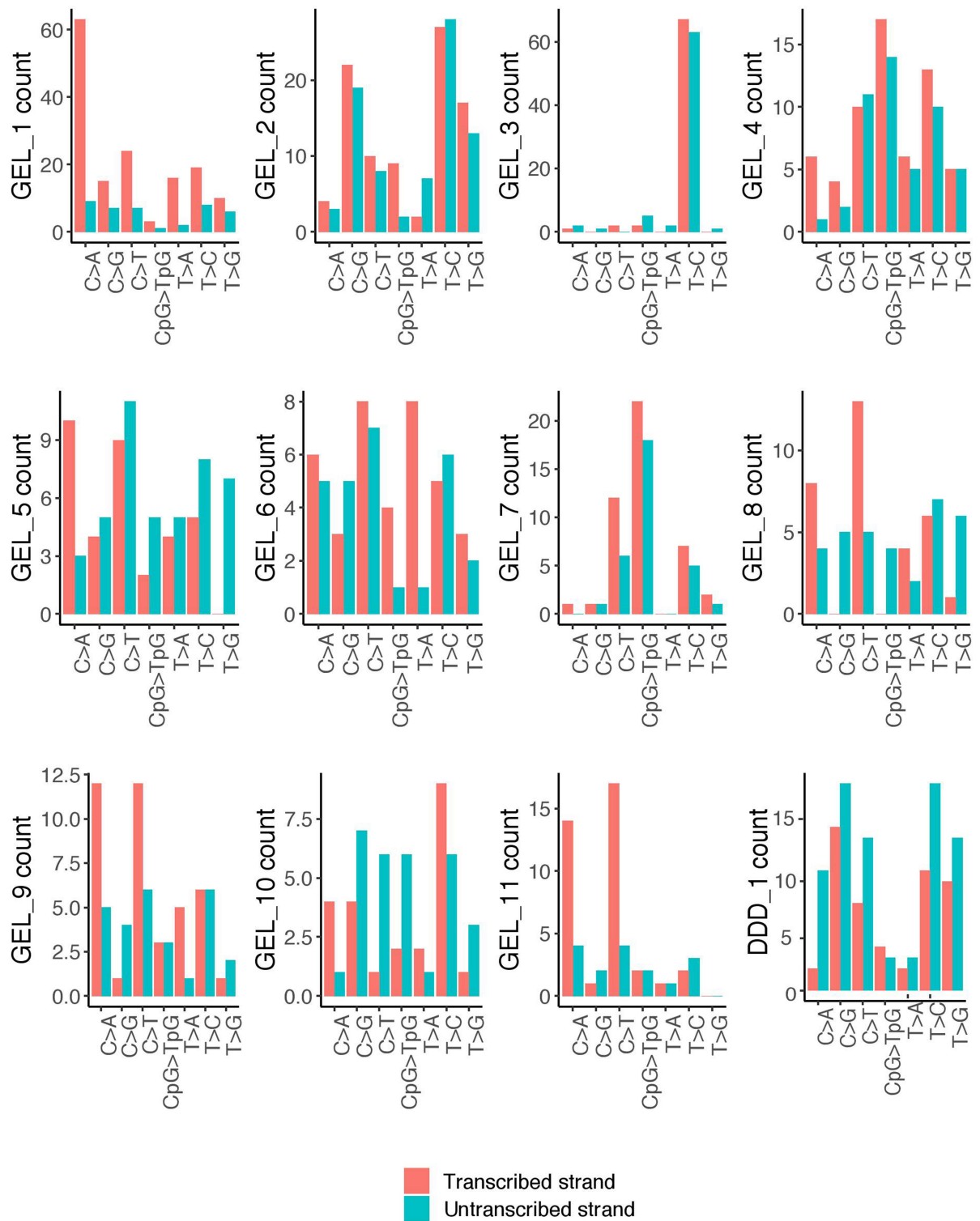

**Extended Data Fig. 6 | Transcriptional strand bias for DNMs in hypermutated individuals.** Plot shows the count of each mutation type on the transcribed and untranscribed strand for each individual. P-values of transcriptional strand bias tests are given in Extended Data Table 1.

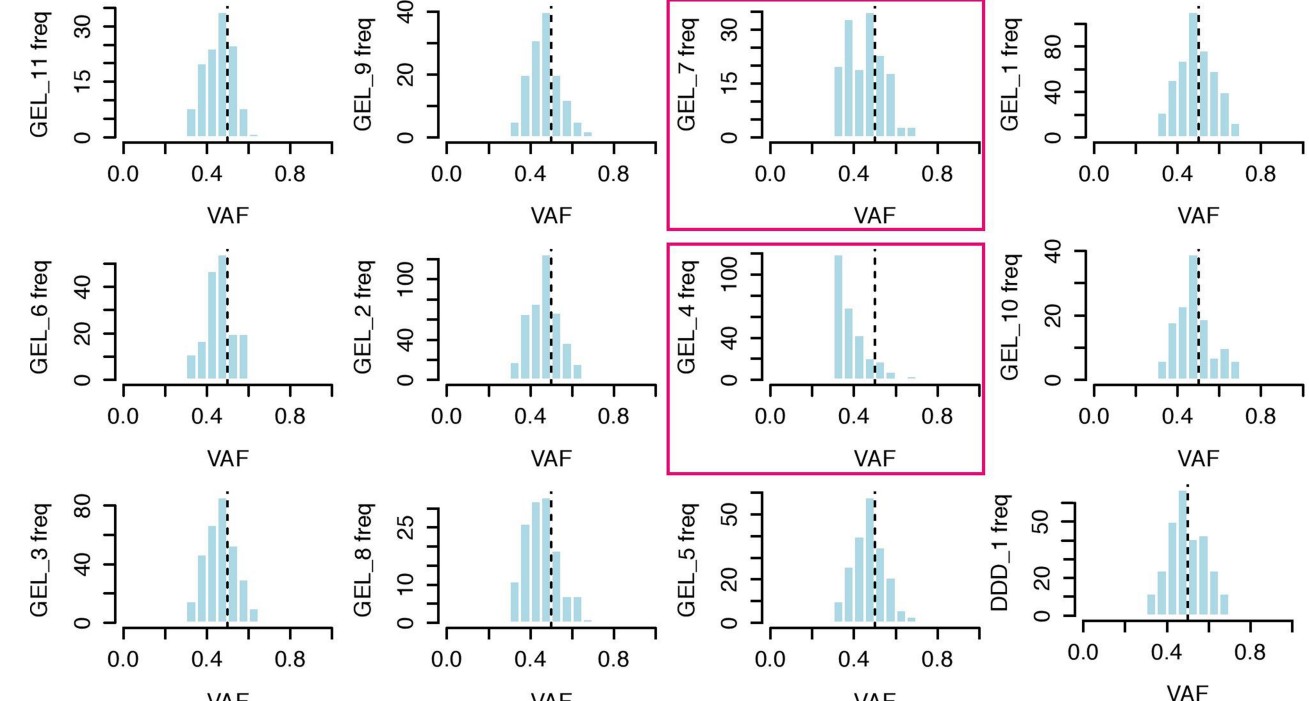

**Extended Data Fig. 7 | Distribution of VAF for DNMs in hypermutated individuals.** The vertical line indicates 0.5 VAF. The two plots highlighted in pink are those where the DNMs appear post-zygotic. P-values of VAF tests are given in Extended Data Table 1.

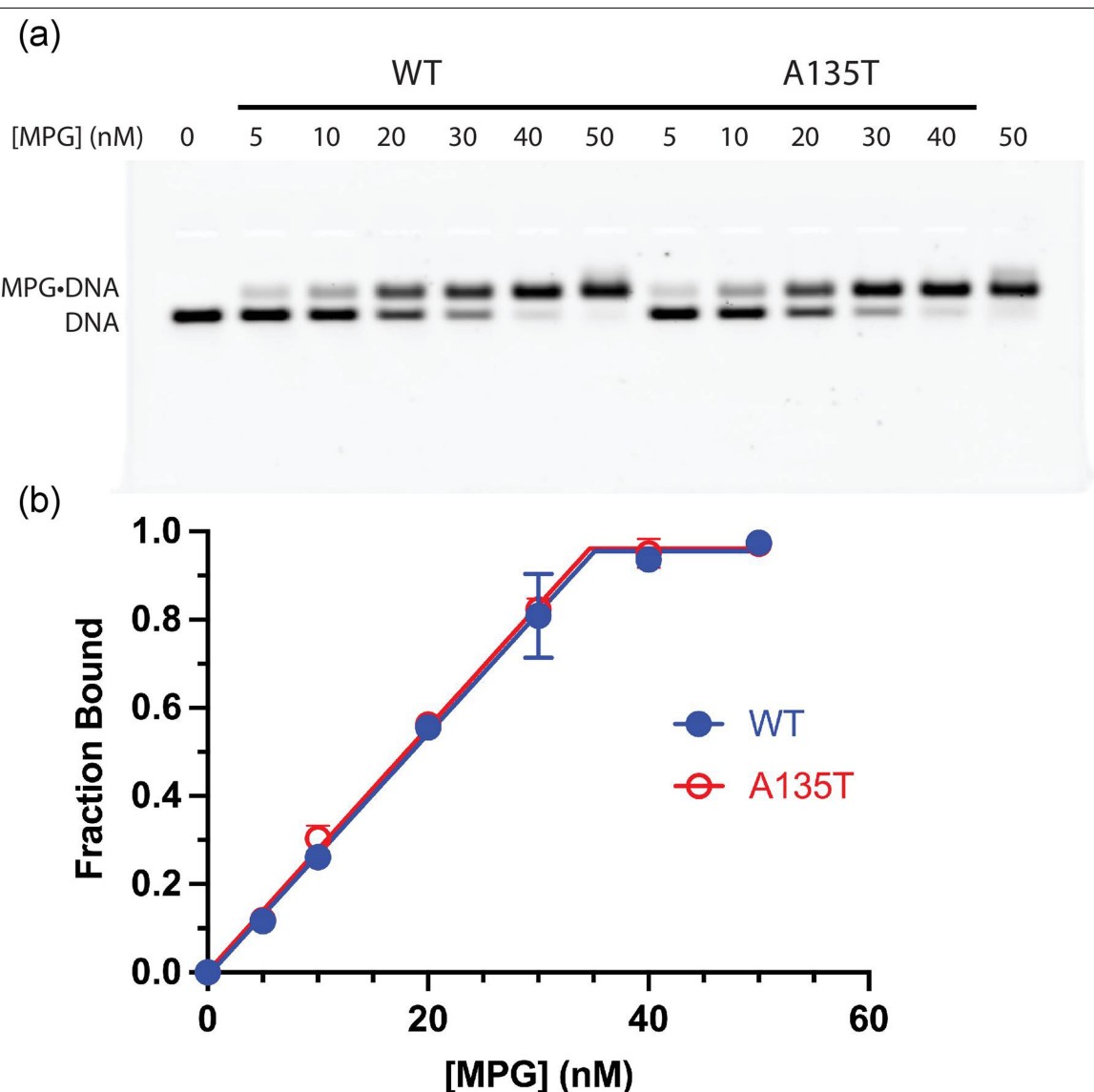

**Extended Data Fig. 8 | Determination of active concentration of *MPG*. (a)** Representative native gel electrophoresis with 20 nM pyrolidine-DNA (Y•T) and varying concentration of WT or A135T *MPG* (25 mM NaHEPES pH 7.5, 100 mM NaCl, 5% v/v glycerol, 1 mM EDTA, 1 mM DTT). Agarose gels (2% w/v) were run in 0.5X TBE buffer at 10 V/cm at 4 °C. **(b)** Independent dilutions were fit to a binding titration to yield an active fraction of 0.57 for both WT and A135T (*n* = 3). This demonstrates that equal concentrations of WT and A135T were tested in the glycosylase assays. The concentrations listed are not corrected by this factor. The points shown are the mean and error bars show 1 standard deviation.

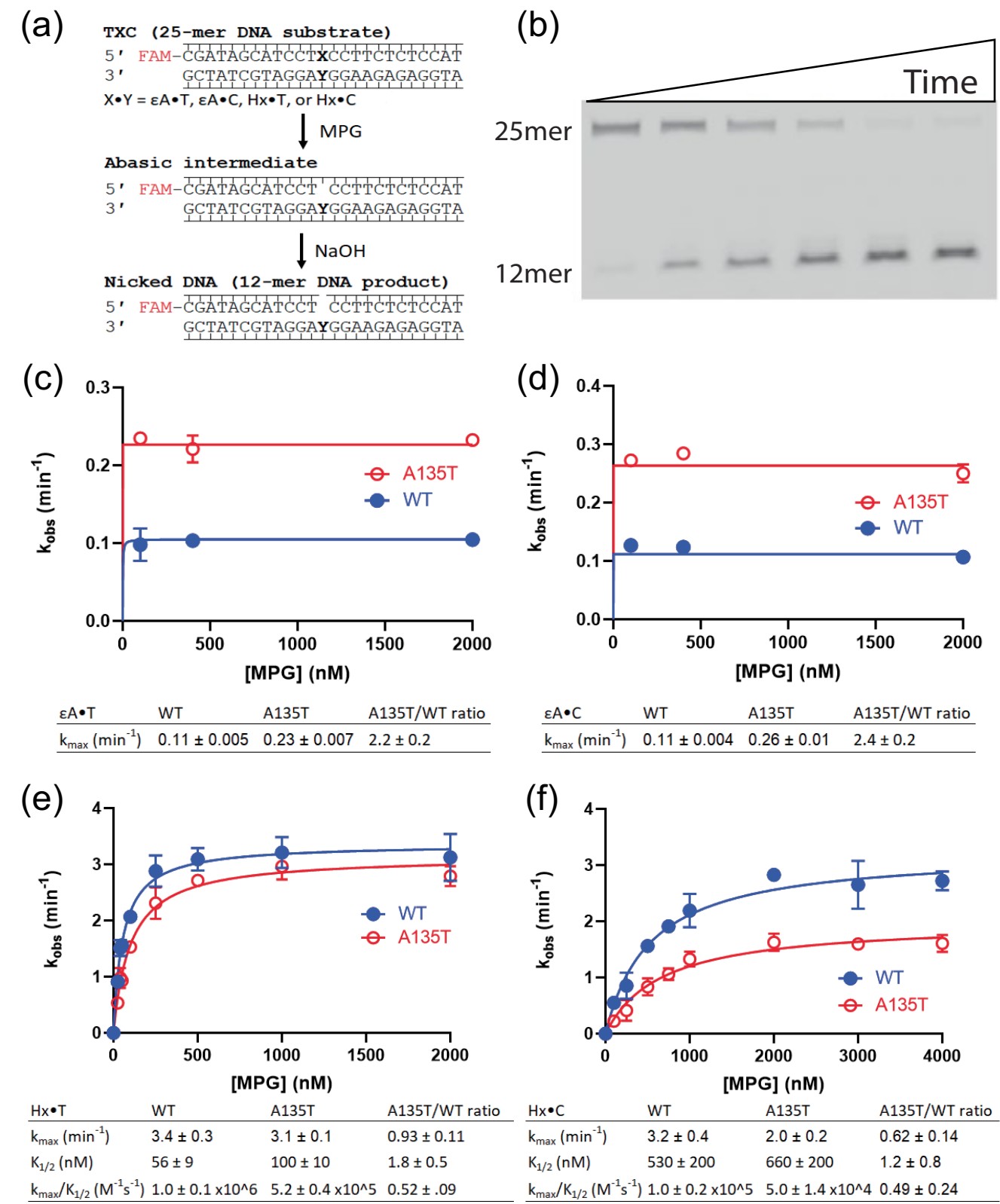

| εA•T | WT | A135T | A135T/WT ratio |
|---|---|---|---|
| $k_{max}$ (min$^{-1}$) | 0.11 ± 0.005 | 0.23 ± 0.007 | 2.2 ± 0.2 |

| εA•C | WT | A135T | A135T/WT ratio |
|---|---|---|---|
| $k_{max}$ (min$^{-1}$) | 0.11 ± 0.004 | 0.26 ± 0.01 | 2.4 ± 0.2 |

| Hx•T | WT | A135T | A135T/WT ratio |
|---|---|---|---|
| $k_{max}$ (min$^{-1}$) | 3.4 ± 0.3 | 3.1 ± 0.1 | 0.93 ± 0.11 |
| $K_{1/2}$ (nM) | 56 ± 9 | 100 ± 10 | 1.8 ± 0.5 |
| $k_{max}/K_{1/2}$ (M$^{-1}$s$^{-1}$) | 1.0 ± 0.1 x10^6 | 5.2 ± 0.4 x10^5 | 0.52 ± .09 |

| Hx•C | WT | A135T | A135T/WT ratio |
|---|---|---|---|
| $k_{max}$ (min$^{-1}$) | 3.2 ± 0.4 | 2.0 ± 0.2 | 0.62 ± 0.14 |
| $K_{1/2}$ (nM) | 530 ± 200 | 660 ± 200 | 1.2 ± 0.8 |
| $k_{max}/K_{1/2}$ (M$^{-1}$s$^{-1}$) | 1.0 ± 0.2 x10^5 | 5.0 ± 1.4 x10^4 | 0.49 ± 0.24 |

**Extended Data Fig. 9** | See next page for caption.

**Extended Data Fig. 9 | In vitro glycosylase activity of WT and A135T *MPG*.**
(a) Glycosylase assay for recombinant protein and 25mer lesion-containing oligonucleotides (O'Brien 2003). *MPG* excises lesion X from X•Y duplex to create an abasic site, which is subsequently hydrolysed by NaOH to create a 12mer product. (b) Representative denaturing gel scanned for fluorescein fluorescence. (c-d) Concentration independent excision of εA from opposing T and C shows increased rate of N-glycosidic bond cleavage by A135T. (panel c, $n = 6$; panel d, $n = 4$) (e-f) Concentration dependence for single-turnover excision of Hx from opposing T and C contexts shows decreased catalytic efficiency for A135T as compared to WT *MPG*. These single turnover rate constants were fit to the equation kobs = kmax [*MPG*]/ (K1/2 + [*MPG*]). (g-h) Steady state concentration dependence for excision of εA was performed in order to measure the catalytic efficiency (kcat/KM) for A135T and WT *MPG* using 5 nM enzyme and the indicated concentration of substrate. To circumvent the tight binding by *MPG*, 800 mM NaCl was added to the standard buffer as previously described, using the equation V/E = kcat/KM[S] (panel e-h, $n = 3$). Mean ± SD is shown for at least 3 independent experiments.

**Extended Data Table 1 | Properties and possible hypermutation sources for germline hypermutated individuals**

| ID | dnSNVs/ dnIndels count | Child age | Paternal age | Maternal age | SNV p-value | Indel p-value | TS bias | Phase (P,M) | Phase p-value | VAF p-value | Potential source of hypermutation |
|---|---|---|---|---|---|---|---|---|---|---|---|
| GEL_1 | 425/16 | 5-10 | 30-35 | 20-25 | 4.2e-90 | 9.4e-05 | 2.1e-40 | 129,1 | 5.3e-13 | 1.00 | Paternal DNA repair defect; homozygous stop-gain *XPC* variant |
| GEL_2 | 375/5 | 10-15 | 25-30 | 25-30 | 2.3e-83 | 0.43 | 0.22 | 106,7 | 6.8e-06 | 0.86 | Paternal chemotherapy; Nephrotic syndrome: Cyclophosphamide, Chlorambucil |
| GEL_3 | 306/4 | 0-5 | 35-40 | 30-35 | 2.5e-44 | 0.73 | 0.86 | 87,5 | 2.3e-05 | 0.89 | Paternal DNA repair defect; homozygous missense *MPG* variant |
| DDD_1 | 276/6 | 6 | 25 | 37 | NA | NA | 3.3e-03 | 72,4 | 9.6e-04 | 1.00 | Paternal chemotherapy; Hodgkins Lymphoma: ABVD, IVE |
| GEL_4 | 262/12 | 10-15 | 30-35 | 20-25 | 1.7e-37 | 0.007 | 0.070 | 36,32 | 6.3e-06 | 3.9e-59 | Post-zygotic hypermutation |
| GEL_5 | 182/8 | 0-5 | 35-40 | 35-40 | 8.4e-14 | 0.19 | 0.15 | 63,4 | 5.8e-04 | 0.88 | Paternal chemotherapy; SLE: drugs unknown |
| GEL_6 | 164/7 | 0-5 | 30-35 | 40-45 | 9.8e-13 | 0.25 | 0.066 | 38,3 | 0.022 | 0.96 | Unknown |
| GEL_7 | 145/9 | 0-5 | 30-35 | 30-35 | 2.4e-09 | 0.08 | 0.02 | 24,16 | 0.012 | 8.3e-04 | Post-zygotic hypermutation |
| GEL_8 | 130/6 | 20-25 | 25-30 | 25-30 | 2.1e-09 | 0.31 | 1.00 | 31,11 | 0.58 | 0.04 | Paternal chemotherapy; Testicular cancer: drugs unknown |
| GEL_9 | 130/5 | 5-10 | 30-35 | 30-35 | 1.2e-07 | 0.53 | 0.016 | 46,2 | 0.0014 | 0.88 | Paternal chemotherapy; Testicular cancer: BEP |
| GEL_10 | 123/5 | 10-15 | 30-35 | 25-30 | 5.3e-08 | 0.48 | 0.082 | 38,0 | 1.1e-04 | 0.81 | Unknown |
| GEL_11 | 110/5 | 10-15 | 25-30 | 25-30 | 8.2e-07 | 0.44 | 6.9e-06 | 28,1 | 0.012 | 0.61 | Paternal chemotherapy; Cancer of long bones, intestinal tract, lung (secondary): Drugs unknown |

Eleven of these individuals were identified in 100kGP as having a significantly large number of dnSNVs (GEL_1-GEL_11) and one hypermutated individual identified in the DDD study (DDD_1). The DNM counts are for autosomal DNMs only. Child age refers to age when sample was taken. Paternal and maternal age refer to age at child's birth. All ages are given as 5 year ranges for 100kGP individuals and the exact age for the DDD individuals. SNV and indel p-value is from testing the number of dnSNVs and dnIndels compared to what we would expect after accounting for parental age. TS bias: transcriptional strand bias poisson two sided p-value for dnSNVS. Phase (P,M): the number of dnSNVs that phase paternally (P) and maternally (M). Phase p-value: from two sided Binomial test for how different this ratio is compared to the observed proportion across all DNMs that phase paternally in 100kGP (~0.78). VAF p-value: one-sided Binomial p-value for testing if number of DNMs with VAF <0.4 is greater than for all DNMs across 100kGP (~0.21). For potential sources of hypermutation when we suspect parental chemotherapy we have detailed the parental cancer and chemotherapy drugs received when relevant. The treatments are abbreviated as follows: BEP (Bleomycin, etoposide and platinum), ABVD (Bleomycin-Dacarbazine-Doxorubicin-Vinblastine) and IVE (Iphosphamide, epirubicin and etoposide).

# nature research

# Reporting Summary

Nature Research wishes to improve the reproducibility of the work that we publish. This form provides structure for consistency and transparency in reporting. For further information on Nature Research policies, see Authors & Referees and the Editorial Policy Checklist.

## Statistics

For all statistical analyses, confirm that the following items are present in the figure legend, table legend, main text, or Methods section.

| n/a | Confirmed | |
|---|---|---|
| ☐ | ☒ | The exact sample size (*n*) for each experimental group/condition, given as a discrete number and unit of measurement |
| ☐ | ☒ | A statement on whether measurements were taken from distinct samples or whether the same sample was measured repeatedly |
| ☐ | ☒ | The statistical test(s) used AND whether they are one- or two-sided<br>*Only common tests should be described solely by name; describe more complex techniques in the Methods section.* |
| ☐ | ☒ | A description of all covariates tested |
| ☐ | ☒ | A description of any assumptions or corrections, such as tests of normality and adjustment for multiple comparisons |
| ☐ | ☒ | A full description of the statistical parameters including central tendency (e.g. means) or other basic estimates (e.g. regression coefficient) AND variation (e.g. standard deviation) or associated estimates of uncertainty (e.g. confidence intervals) |
| ☐ | ☒ | For null hypothesis testing, the test statistic (e.g. $F$, $t$, $r$) with confidence intervals, effect sizes, degrees of freedom and $P$ value noted<br>*Give P values as exact values whenever suitable.* |
| ☒ | ☐ | For Bayesian analysis, information on the choice of priors and Markov chain Monte Carlo settings |
| ☒ | ☐ | For hierarchical and complex designs, identification of the appropriate level for tests and full reporting of outcomes |
| ☐ | ☒ | Estimates of effect sizes (e.g. Cohen's *d*, Pearson's *r*), indicating how they were calculated |

*Our web collection on statistics for biologists contains articles on many of the points above.*

## Software and code

Policy information about availability of computer code

| Data collection | NA |
|---|---|
| Data analysis | Analyses were performed primarily in R (4.0.1). Phasing of mutations was performed with a custom Python (3) script available at https://github.com/queenjobo/PhaseMyDeNovo. Signature extraction was performed using SigProfiler (v1.0.17). Details of software used for sequence alignment, variant calling and de novo mutation calling are given in the Methods. Analysis of gels was done with ImageQuant TL 7.0 (Cytiva). |

For manuscripts utilizing custom algorithms or software that are central to the research but not yet described in published literature, software must be made available to editors/reviewers. We strongly encourage code deposition in a community repository (e.g. GitHub). See the Nature Research guidelines for submitting code & software for further information.

## Data

Policy information about availability of data

All manuscripts must include a data availability statement. This statement should provide the following information, where applicable:

- Accession codes, unique identifiers, or web links for publicly available datasets
- A list of figures that have associated raw data
- A description of any restrictions on data availability

Sequence and variant-level data and phenotypic data for the DDD study data are available from the European Genome-phenome Archive (EGA; https://www.ebi.ac.uk/ega/) with study ID EGAS00001000775. This is under managed access to ensure that the work proposed by the researchers is allowed under the study's ethical approval.
Sequence and variant-level data (including the de novo mutations dataset) and phenotypic data from the 100,000 Genomes Project can be accessed by application to Genomics England Ltd following the procedure outlined at: https://www.genomicsengland.co.uk/about-gecip/joining-researchcommunity/
Genome Aggregation Database (gnomAD v2.1.1; https://gnomad.broadinstitute.org/)

# Field-specific reporting

Please select the one below that is the best fit for your research. If you are not sure, read the appropriate sections before making your selection.

☒ Life sciences ☐ Behavioural & social sciences ☐ Ecological, evolutionary & environmental sciences

For a reference copy of the document with all sections, see nature.com/documents/nr-reporting-summary-flat.pdf

# Life sciences study design

All studies must disclose on these points even when the disclosure is negative.

| | |
|---|---|
| Sample size | This is an observational study: the sample size is 7,930 exome sequenced parent-child trios from the DDD study and 13,949 whole genome sequenced parent-child trios from the 100,000 Genome Project. No power calculations were done prior to analyses; we used all available samples. |
| Data exclusions | We excluded 12 trios in the 100,000 Genomes Project with a high false positive rate of de novo mutations as outlined in the Methods. |
| Replication | There was no replication in this study as it was focused on specific outliers in an observational study which is unable to be replicated. However these outliers were identified in two separate studies. Analyses performed across the whole cohort were not replicated although results were compared and found to be very similar to previous published results from other studies. |
| Randomization | There was no randomization in this study as it was not applicable since we were identifying specific outliers in an observational study. |
| Blinding | There was no blinding in this study as it was not applicable as it was an observational study focussed on genetic data. |

# Reporting for specific materials, systems and methods

We require information from authors about some types of materials, experimental systems and methods used in many studies. Here, indicate whether each material, system or method listed is relevant to your study. If you are not sure if a list item applies to your research, read the appropriate section before selecting a response.

### Materials & experimental systems

| n/a | Involved in the study |
|---|---|
| ☒ | ☐ Antibodies |
| ☒ | ☐ Eukaryotic cell lines |
| ☒ | ☐ Palaeontology |
| ☒ | ☐ Animals and other organisms |
| ☐ | ☒ Human research participants |
| ☒ | ☐ Clinical data |

### Methods

| n/a | Involved in the study |
|---|---|
| ☒ | ☐ ChIP-seq |
| ☒ | ☐ Flow cytometry |
| ☒ | ☐ MRI-based neuroimaging |

## Human research participants

Policy information about studies involving human research participants

| | |
|---|---|
| Population characteristics | The Deciphering Developmental Disorders study consists of individuals with severe developmental disorders recruited along with their parents. The rare disease arm of the 100,000 Genomes Projects consists of individuals with rare disease (that fall into 15 rare disease domains) recruited along with their parents (see https://www.nature.com/articles/s41586-020-2434-2#Sec21 for details) |
| Recruitment | Recruitment differed for the two cohorts:<br><br>Deciphering Developmental Disorders (DDD): Patients with severe, undiagnosed developmental disorders were recruited from 24 regional genetics services within the United Kingdom National Health Service and the Republic of Ireland. These analyses involve 7,930 trios who have been analyzed in previous publications. Patients typically had some prior genetic testing (e.g. an array or a single gene test) before recruitment into the study.<br><br>100,00 Genomes Project: Study participants were enrolled by one of three mechanisms between December 2012 and March 2017 under the overall coordination of the National Institute for Health Research BioResource (NBR) at Cambridge University Hospitals. Patients with rare diseases and their close relatives were enrolled into 15 rare disease domains approved by the Sequencing and Informatics Committee of the NBR. Participants in the rare disease domains were recruited mainly at NHS Hospitals in the United Kingdom, but also at hospitals overseas. |

Ethics oversight

DDD: The study was approved by the UK Research Ethics Committee (10/H0305/83 granted by the Cambridge South Research Ethics Committee, and GEN/284/12 granted by the Republic of Ireland Research Ethics Committee).

100,000 Genomes Project: All participants provided written informed consent, either under the East of England Cambridge South national research ethics committee (REC) reference no. 13/EE/0325 or under ethics for other REC-approved studies.

Note that full information on the approval of the study protocol must also be provided in the manuscript.

