## [Peer Review File · Nature]

Manuscript Title: Genetic and chemotherapeutic influences on germline hypermutation

Reviewer Comments & Author Rebuttals

Reviewer Reports on the Initial Version:

Referees' comments:

Referee #1 (Remarks to the Author):

Kaplanis et al examined whole-genome sequencing datasets collected from more than 20,000 family trios from two developmental disorder cohorts (DDD and UK100K) to quantify germline mutation loads in the offspring of each trio. Their examination of phased, de novo single-nucleotide variants identified 12 individuals (9/12 of paternal origin) harboring a germline hypermutation phenotype having a > 2-fold adjusted mutation rate compared to other subjects included in the analysis. The team investigated three causes of the observed germline hypermutation: 1) paternal defects in DNA repair genes, 2) paternal exposure to chemotherapeutic agents, and 3) post-zygotic mutational factors. Overall, the manuscript is clearly written and demonstrates an expected result - that germline hypermutation is rare. Of particular interest, however, is the observation that individuals with hypermutated germlines lack genetic disease. We have some concerns and suggestions to improve the clarity of the manuscript.

Major:

The authors state that they found parental age effects in phased mutations were not significantly different from unphased results. It is unclear how they correlated mutations to parental age effects in the absence of phasing? Do you mean parental age with total mutation load? The manuscript should clarify.

The manuscript refers to multiple mutation signatures (e.g., signature 1 and 5) without giving the reader a clearer sense of what these abstract signatures "mean". Give examples of the dominant mutation types and/or the proposed underlying mechanisms. Otherwise, even readers familiar with the concept of these abstract signatures will be left wanting a better understanding of the biology. For example, how do the patterns in Figure 1b connect to signatures 1 and 5? Could that be better conveyed in the figure?

Does the assay really confirm "that the A135T substitution alters the MPG binding pocket" when it seems that the authors only measured the efficiency of DNA damage repair pathways. Could the mutation impact the gene in another way? It seems that Supplemental Fig 8 says that the mutated MPG binds to DNA similarly to WT. Perhaps the MPG mutation is not affecting binding affinity?

Elevated germline mutation rates could explain the onset of testicular cancer in GEL_8/9. It seems that one cannot conclude that cisplatin alone is causal for elevated germline mutation rates. In particular, the authors could provide a discussion of their conclusion in light of the blood-testis

barrier and how chemotherapies interact with it.

Germline mutation rates in testicular cancer patients could have been elevated prior to chemotherapy

<https://pubmed.ncbi.nlm.nih.gov/8978597/>: looked at how cisplatin can disrupt the relationship between germinal and Sertoli cells

<https://www.sciencedirect.com/science/article/pii/S1470204502007763?via=ihub>: discuss clinical relevance of cisplatin therapies in treating testicular tumors and infertility

https://www.readcube.com/articles/10.1038/ncponc0018?no_publisher_access=1&r3_referer=nature&referrer_host=www.nature.com: discuss the importance of BTB in normal testicular health and fertility

In the discussion, it would be helpful to comment that longitudinal analyses of multiple offspring were not conducted when discussing how the observed germline mutation counts/rates compare to what has been found in the general population.

The authors state that “parental age explained 70% of the variance in numbers of dnSNVs per offspring --- smaller than a previous estimate of 95% based on a sample of 78 families. They talk about re-sampling of 78 trios from 100,000 GP to estimate parental age effect on dnSNV rates (include this in the results section?). Perhaps to bolster their claim that their findings are more “reliable” given their large sample size, authors should expand on how their findings are similar/different to what has been previously observed. As described the argument is not terribly clear.

Minor:

Sup Fig 2a refers to indels and 2b refers to SNVs which is the opposite as stated in the paper

"We selected a subset of trios from the DDD study where the offspring had an unusually high number of exonic DNMs given their parental". How many were sequenced? To what depth? How did you define unusual? These important details could be succinctly brought forth to the main text.

SBSHYP is not mentioned explicitly in the main text, only referred to as a “novel mutational signature”

Can you test for increased germline muta rate at STR loci in GEL_1 (mutated XPC) from WGS data, similar to what has been done in mice?

What cell lines were used to determine the impact of observed MPG A135T mutation?

Axis labels on Fig 2b are very difficult to read.

What was the fold-change increase in germline hypermutation in GEL4/7?  this is mentioned in the other sections but is neglected in this section

Unclear from the manuscript body which diseases were included

Unclear what coverage was used in the initial cohorts

Supplemental Fig 3, 5, 6 does not have a caption

Supplemental figures are out of order

“Heterozygous PTVs in known somatic mutator genes do not always have a similar effect in the germline”. This may be a bit strong since only one gene (MBD4) was tested among n=13 paternal carriers of MBD4 PTVs

Aaron Quinlan (mentor) and Jason Kunisaki (graduate student)

Referee #2 (Remarks to the Author):

The manuscript by Kaplanis et al entitled “Genetic and pharmacological causes of germline hypermutation” is well written and straightforward to follow. The authors analyzed genome wide sequences from 21,879 families with rare genetic diseases and identified 12 hypermutated individuals with excessive numbers of de novo single nucleotide variants and small indels after correcting for paternal age and sequencing quality. This is a large data set and the study is straight forward. The authors do a nice job of phasing the variants and characterizing the mutation signatures. This is important to understand mechanism for the hypermutability phenotype. The authors also do a nice job of more precisely estimating the paternal age effect which appears to account for slightly less of the variance than the previous estimates based on a much smaller sample set. The results however are largely negative and obvious. Because the results are negative, they are largely reassuring that the absolute frequency of hypermutation is low. That risk may be slightly higher in consanguineous communities, but even there would be quite low. There is also largely a negative result to suggest that environmental carcinogens are a major contributor to hypermutation.

Of the 12 individuals who are hypermutated, six of them had fathers who received chemotherapy prior to conception. This is well known clinically that chemo is associated with increased frequency of mutations and is one of the reasons why we often bank gametes prior to chemotherapy.

There are two interesting cases of fathers with bilallelic recessive variants in DNA repair genes. One father is clinically affected with xeroderma pigmentosa due to homozygous nonsense variant in XPC associated with nucleotide excision repair. It is not surprising that he would have gametes that are hypermutated. A second father has a rare homozygous missense variant in MPG. This is more interesting since MPG has not been described as a cancer susceptibility gene previously although the gene encodes a gene that is involved in the initiation of base excision repair pathway.

There are interesting confounds by indication for this analysis. Although several of the fathers received chemotherapy, this was also because they had cancer at a young age. I would be very careful to check that there are no germline cancer susceptibility genes although I appreciate that the authors think they did this. For example, the father of GEL_11 was diagnosed with three separate cancers at a young age. Osteosarcoma, lung cancer, and cancer of the intestinal tract is quite an unusual spectrum, especially for such a young individual. I would look more carefully at that father for some sort of heritable cancer syndrome or environmental exposure.

Somatic mutations in the child are rare in young children but increase with age. It would be helpful to list the age of the children in table 1.

For the GE L_7 child, it is not clear to me if the child has the TP53 variant observed to be mosaic in the mother. If so, the variant is germline and maybe a contributor to that hypermutation.

Because there can be technical issues with these variant calls, I recommend that the authors confirm the presence of a representative sample of the variants in the hypermutable cases. This is especially important for the cases of unknown source of hypermutation.

A minor point: I would add a few fields to table 1: the type of chemotherapy received by the father if it is known and the type of cancer in the father if it is known.

Please provide the list of DNA repair genes used in this analysis and how did you select them?

I would replace pharmacological in the title to specify that the drugs they examined were chemotherapeutic agents.

Referee #3 (Remarks to the Author):

Summary of the study and key findings:

Kaplanis and colleagues have conducted a very interesting and important study into the causes of germline hypermutation and identified novel contributing factors, including paternal exposure to chemotherapy prior to conception and paternal genetic variants in DNA repair genes. This study is much larger than previous analyses, utilizing two cohorts of parent-offspring trios totaling ~20,000 families with children that have developmental disorders or rare diseases.

The authors confirm, as shown previously in smaller studies, that the majority of germline mutations are explained by parental age, particularly of the father. They were able to phase ~25% (225,854/921,433) de novo mutations (DNMs) and found that 77% of these were paternal in origin. The majority of both maternal and paternal mutations can be explained by COSMIC signatures SBS1 and 5.

The authors went on to identify 12 individuals hypermutated in the germline (after accounting for parental age), harbouring ~2-7-fold more DNMs than the median number of DNMs/individual in the 100kGP cohort. They assessed the mutational spectra of the DNMs in each individual and used SigProfiler to extract mutational signatures. These mutational signatures were then used to identify the potential causative mechanism(s) for the hypermutation.

The mutational signatures in two patients indicated a contribution of DNA repair deficiency. In both cases, the fathers were found to carry homozygous nonsynonymous variants in DNA repair genes. In one case (GEL_1) the father had a rare nonsense variant in XPC, a key component in the nucleotide

excision repair pathway, and the child had mutations which match signature SBS8, associated with NER-deficiency. While XPC deficiency is known to increase the risk of developing cancer, the link to human germline mutagenesis is novel. In the second case (GEL_3), a contribution of SBS26 is observed, which is associated with defective mismatch repair. The father has a missense variant (A135T) in the gene MPG, which codes for a glycosylase involved in the repair of alkylated and deaminated purines. Using in vitro functional assays, the authors show that this variant changes the activity of MPG, but further work would be required to establish a link between the A135T variant and SBS26.

Significantly, the authors also establish a link between germline mutations and paternal exposure to chemotherapy. SBS31, a signature associated with platinum-drug exposure, contributed to the DNM mutational spectra of three children. The fathers of these children had a history of cancer and chemotherapy prior to conception, implicating platinum-based treatment as a source of germline mutagenesis. Two other children shared paternally-biased DNM mutational spectra, from which the authors extracted a novel mutational signature named SBSHYP. As both fathers received treatment with nitrogen mustard alkylating agent chemotherapy prior to conception, these agents are suspected to be the cause of the novel signature.

Finally, the high level of DNMs in two further individuals was found to be caused by post-zygotic hypermutation, and consisted of SBS1 and 5.

General comments on the study and manuscript:

This is a well written article, clearly presented and accessible. The design of the study, the methods used and the interpretation of the results appear to be sound. Analyses were well thought-out and thorough, documented in clear Figures and Tables. Statistical tests and probability values are described for each analysis. Conclusions drawn from the results are valid and not overstated. The reference of previous literature is appropriate and relevant.

The work presented here builds upon previous studies by utilising a larger dataset of parent-offspring trios. This study confirms previous findings (e.g. the proportion of DNMs attributable to the father and mother) but also provides important, new insight by identifying causative elements for the mutational signatures found in germline hypermutated individuals (e.g. paternal chemotherapy exposure), which has not been shown previously. The authors also highlight that germline hypermutation is ultimately rare. The findings in this study will certainly be of interest to a diverse audience, spanning the fields of human genetics, DNA repair and genetic toxicology and will stimulate further research.

Specific comments for discussion or improvement:

1. Fig 1b

- Please lighten the purple colour as it is difficult to read black numbers on dark purple.

2. Fig 2a

- It may be useful to some readers for the classification of each of these signatures to be briefly labeled on the Figure or in the legend (ie SBS31: platinum drug; SBS26: MMR-deficiency).

- Supplemental Figure 13 indicates that one would expect some contribution of SBS5 in every individual, as this signature is detected in all of the controls, as well as the non-hypermethylated individuals with parents who received chemotherapy. Therefore, could the authors explain, or discuss, why there is no contribution of SBS5 in GEL_2 or GEL_11?

3. In the Results section “Paternal defects in DNA repair”, the authors show that GEL_3 harbours mutations that map to SBS26, which is linked to defective MMR, and the father has a variant in the gene MPG. Could the authors comment on other MPG variants present in humans? For example, I found at least one study describing human variants in MPG:

<https://doi.org/10.1074/jbc.M114.627000>. Some in vitro work was done by the authors to show that the activity of MPG may be affected by the A135T variant. The variant had decreased activity for one adduct and increased activity for another. Could the authors put these results into context, as it is not clear how significant a 2-fold decrease or increase would be. Have any other variants for MPG been assessed in the in vitro assay used by the authors, and if so, how did those results compare with the results presented for the A135T variant?

4. In the Results section “Parental treatment with chemotherapy prior to conception”, it would be useful to briefly describe the type of DNA damage and mutations caused by cisplatin, rather than simply referring to SBS31 (especially since SBS31 is not illustrated anywhere in the manuscript). Further, could the authors comment on whether there is any evidence in the literature that platinum drugs induce germline mutation in animal models?

5. Supplemental Figure 2: The y-axis labeling needs to be fixed in (b).

6. Supplemental Figure 5: In the lefthand panel, many of the graphs have bars that are higher than the y-axis. Please check and amend. For example, see e) and h).

7. Supplemental Figure 10: Again, check the height of the y-axis, as some bars overtake the axis label. For example, see MBD4_9 and MBD4_13.

8. Supplemental Figure 12: Please improve the resolution of this figure as it is almost impossible to read the text when zoomed in.

Author Rebuttals to Initial Comments:

Genetic and chemotherapeutic causes of germline hypermutation

2021-05-08670

We would like to thank the reviewers for the expert evaluation of our manuscript and constructive feedback. Overall, we are very pleased that the reviewers have recognised the significance of our work. The reviewers' comments helped to improve the quality of our data interpretation, and enhanced the clarity of the manuscript.

While revising our manuscript we discovered an error in the filtering step of DNMs that impacted ~5% of the 100,000 Genomes Project dataset. All GRCh38 DNMs had been filtered using GRCh37 coordinates of repeat regions. This issue is now fixed and we have manually inspected 500 randomly sampled DNMs and have reestimated the true positive rate as 97% for dnSNVs and 92% for dnIndels which is slightly higher than before. All of the main findings remained the same and there were a few slight changes:

- The heritability estimate of the paternal germline mutation rate is no longer significant ($p = 0.095$) although the actual heritability estimate is very similar ($h^2 = 0.5$).
- We additionally identified the presence of an extra mutational signature SBS24 in GEL_5, and have added a sentence about this in the main text.
- DNMs phase paternally significantly ($p < 0.05/12$) more than we expect in 8 out of the 12 individuals (9/12 previously). This is because the p-value just crosses over the significance threshold for GEL_9 ($p = 0.0044$).

We have detailed our response to each point below, the reviewer comments are in black, with our responses in blue.

Referees' comments:

Referee #1 (Remarks to the Author):

Kaplanis et al examined whole-genome sequencing datasets collected from more than 20,000 family trios from two developmental disorder cohorts (DDD and UK100K) to quantify germline mutation loads in the offspring of each trio. Their examination of phased, de novo single-nucleotide variants identified 12 individuals (9/12 of paternal origin) harboring a germline hypermutation phenotype having a > 2-fold adjusted mutation rate compared to other subjects included in the analysis. The team investigated three causes of the observed germline hypermutation: 1) paternal defects in DNA repair genes, 2) paternal exposure to chemotherapeutic agents, and 3) post-zygotic mutational factors. Overall, the manuscript is clearly written and demonstrates an expected result - that germline hypermutation is rare. Of particular interest, however, is the observation that individuals with hypermutated germlines lack genetic disease. We have some concerns and suggestions to improve the clarity of the manuscript.

Major:

1.1 The authors state that they found parental age effects in phased mutations were not significantly different from unphased results. It is unclear how they correlated mutations to parental age effects in the absence of phasing? Do you mean parental age with total mutation load? The manuscript should clarify.

The reviewer is correct, we mean paternal age with total mutation load. We have now clarified this in the manuscript.

1.2 The manuscript refers to multiple mutation signatures (e.g., signature 1 and 5) without giving the reader a clearer sense of what these abstract signatures "mean". Give examples of the dominant mutation types and/or the proposed underlying mechanisms. Otherwise, even readers familiar with the concept of these abstract signatures will be left wanting a better understanding of the biology. For example, how do the patterns in Figure 1b connect to signatures 1 and 5? Could that be better conveyed in the figure?

We thank the reviewer for their suggestion that we have now defined these signatures in the manuscript. We have added the following in the manuscript (lines 48-50):

The majority of germline mutation can be explained by two of these signatures, termed signature 1 (SBS1), likely due to deamination of 5-methylcytosine²⁰, and signature 5 (SBS5), thought to be a pervasive and relatively clock-like endogenous process. Both signatures are ubiquitous among normal and cancer cell types^{21,22} and have been reported previously in trio-studies¹³.

1.3 Does the assay really confirm "that the A135T substitution alters the MPG binding pocket" when it seems that the authors only measured the efficiency of DNA damage repair pathways. Could the mutation impact the gene in another way? It seems that Supplemental Fig 8 says that the mutated MPG binds to DNA similarly to WT. Perhaps the MPG mutation is not affecting binding affinity?

We apologise that the concise explanation and interpretation of these experiments in the manuscript was not clearer.

Quantitative biochemical study of the different MPG variants requires accurate determination of the concentration of active sites. Supplementary Figure 8 shows binding to a pyrrolidine abasic DNA mimic in order to precisely measure the amount of active enzyme. This additional assay is required because neither ethenoA nor hypoxanthine are substrates for which we can easily measure direct binding (ethenoA is too tight binding and hypoxanthine is too quickly hydrolysed). The fact that mutated and wildtype forms of MPG are indistinguishable in Supplementary Figure 8 indicates that active site concentrations are being accurately determined, and that the quantitative differences between the mutated and wildtype forms of MPG shown in the glycosylase assays are not due to inaccurate measurement of active site concentrations. We have now revised the legend to Supplementary Figure 8 to make this clearer to a broader range of readers.

We are not suggesting that A135T affects binding affinity per se, but rather changes the specificity of the enzyme by causing rearrangement in the active site pocket (catalytic activity requires both binding and transition-state stabilization and these cannot be readily distinguished). Note that our experiments show that one lesion is more efficiently excised in the mutant form of MPG, and the other is more poorly excised. We do not yet know the structural details for how the pocket has changed, but our data shows that the specificity is altered.

In the original manuscript, we reported comparisons of the catalytic efficiency (kcat/KM) of mutant and wildtype MPG only for the hypoxanthine substrate. We have now generated additional data on the catalytic efficiency for the ethenoA lesion. These data have been incorporated as two additional panels (g and h) in Supplementary Figure 9 of the revised manuscript. kcat/KM measures changes in how the substrate is bound in the transition state and these data show changes in specificity between different types of substrate. These additional data continue to support the conclusions in the original manuscript, that the catalytic efficiency of the A135T variant and wildtype forms of MPG are different, and that the effects for the two lesions investigated are in different directions (increased and decreased). We have added additional references into the legend for Supplementary Figure 9 to cite the previous use and validation of the methods used and to support the methodological adaptation necessary to circumvent the tight binding of MPG to the ethenoA lesion such that the catalytic efficiencies could be determined.

Please note that in our response to Reviewer 3 below (3.4) we have also added a supplementary table that compares the A135T data to 14 other MPG variants that have been characterised previously. This table shows that the A135T variant causes a greater increase in single turnover rate for the ethenoA substrate than any previously characterised variants, which includes variants that have been previously determined to cause a mutator phenotype when introduced into yeast.

1.4 Elevated germline mutation rates could explain the onset of testicular cancer in GEL_8/9. It seems that one cannot conclude that cisplatin alone is causal for elevated germline mutation rates. In particular, the authors could provide a discussion of their conclusion in light of the blood-testis barrier and how chemotherapies interact with it.

Germline mutation rates in testicular cancer patients could have been elevated prior to chemotherapy

<https://pubmed.ncbi.nlm.nih.gov/8978597/> [pubmed.ncbi.nlm.nih.gov]: looked at how cisplatin can disrupt the relationship between germinal and Sertoli cells

<https://www.sciencedirect.com/science/article/pii/S1470204502007763?via=ihub> [[sciencedirect.com](https://www.sciencedirect.com/)]: discuss clinical relevance of cisplatin therapies in treating testicular tumors and infertility

https://www.readcube.com/articles/10.1038/ncponc0018?no_publisher_access=1&r3_referer=nature&referrer_host=www.nature.com [[readcube.com](https://www.readcube.com/)]: discuss the importance of BTB in normal testicular health and fertility

To be clear, it is the fathers of the hypermutated individuals GEL_8 and GEL_9 who have had testicular cancer, and not GEL_8 and GEL_9 themselves. As described in the supplementary table, GEL_8 has a disorder of copper homeostasis and GEL_9 has intellectual disability. The contribution of SBS31 in both GEL_8 and GEL_9 strongly suggests that cisplatin is the primary cause for the elevated mutation rates in these two individuals. This signature has been associated with cisplatin and observed in several tumour and human cell lines (Boot et al. 2018; Pich et al. 2019). The number of DNMs contributed by SBS1 and SBS5, expected signatures of germline mutation, are not significantly increased in these individuals. Removing the contribution of SBS31 these individuals would have 100 (GEL_8) and 73 (GEL_9) dnSNVs. Exact pathways that cisplatin affects germline cells within the seminiferous tubules are not clear.

Following the reviewer's suggestion, we have included mention of the importance of the blood-testis barrier when considering the germline mutagenic effects of different chemotherapies in the Discussion (line 467), citing the relevant review suggested by the reviewer.

1.5 In the discussion, it would be helpful to comment that longitudinal analyses of multiple offspring were not conducted when discussing how the observed germline mutation counts/rates compare to what has been found in the general population.

We are not quite clear what the reviewer is suggesting here. Is the reviewer suggesting sampling the same individual at multiple timepoints or analysing siblings that are conceived from the same parents at different times? The mutation counts/rate reported here is in keeping with the estimates from analysis of multiple offspring families [eg. Sassani et al. 2019 and Rahbari et al. 2016]. Hence, it is not clear to us how further discussion on this topic might further assist the readers of this work.

The authors state that "parental age explained 70% of the variance in numbers of dnSNVs per offspring --- smaller than a previous estimate of 95% based on a sample of 78 families. They talk about re-sampling of 78 trios from 100,000 GP to estimate parental age effect on dnSNV rates (include this in the results section?). Perhaps to bolster their claim that their findings are more "reliable" given their large sample size, authors should expand on how their findings are similar/different to what has been previously observed. As described the argument is not terribly clear.

We have moved this analysis to the Results section now and expanded on our findings, which hopefully make our reasoning clearer..

We hypothesised that due to the (100X) smaller sample size of the previous study, the apparent discordance in variance explained by paternal age could be explained simply by (stochastic) uncertainty in the estimate in the previous, smaller, study. To test this hypothesis, we randomly sampled 78 families from the 100,000 GP many times and estimated the variance explained for each. We observed that there was considerable stochastic variation in the point estimate of

the variance explained between these sub-samples. The 95% interval of these estimates (52-100%) included that both the prior (95%) and current (70%) estimates of the variance explained by parental age. We do not think any differences in specificity of DNM calling between the two studies would be sufficient to explain the difference since including various data quality measures in our regression model only explain a small fraction of variance (~1.5%).

From a purely statistical perspective, one would certainly expect that an estimate of variance explained based on a much larger sample size would have considerably greater precision than one based on a much smaller sample size. In theory, we could demonstrate this empirically by sub-sampling larger sample sizes and showing that the 95% confidence interval of these estimates shrinks, but we think this is sufficiently self-evident.

Minor:

1.6 Sup Fig 2a refers to indels and 2b refers to SNVs which is the opposite as stated in the paper

This has now been corrected.

1.7 "We selected a subset of trios from the DDD study where the offspring had an unusually high number of exonic DNMs given their parental". How many were sequenced? To what depth? How did you define unusual? These important details could be succinctly brought forth to the main text.

We have amended the main text as follows to include these details:

We selected nine trios from the DDD study with the largest number of exonic DNMs in the offspring, given their parental ages i , which were subsequently whole genome sequenced at >30X coverage to characterise DNMs genome-wide.

1.8 SBSHYP is not mentioned explicitly in the main text, only referred to as a "novel mutational signature"

This is now mentioned explicitly in the main text. (line 255)

Can you test for increased germline muta rate at STR loci in GEL_1 (mutated XPC) from WGS data, similar to what has been done in mice?

Thank you for this suggestion. This cannot be tested using existing data and would require additional algorithms to be applied across the 100kGP cohort to construct a null for the STR mutation rate which is beyond the scope of the study. GEL_1 is the only hypermutated individual that also has a significantly increased number of *de novo* indels which is indicative of replication slippage. Having examined these 16 indels we observed that 11 of these occur in an (A), (AT) or (AG) context..

1.9 What cell lines were used to determine the impact of observed MPG A135T mutation?

The functional characterisation of MPG was performed *in vitro* and not in a cellular assay, as described in the Methods, with the catalytic domain of the protein expressed in BL21(DE3) Rosetta2 *E. coli* and then purified. Examining the *in vivo* mutagenic consequences of the A135T variant of MPG, is of interest, but is beyond the scope of this study.

1. 10 Axis labels on Fig 2b are very difficult to read.

These have now been made larger.

1.11 What was the fold-change increase in germline hypermutation in GEL4/7? -
-> this is mentioned in the other sections but is neglected in this section

The fold change is ~4 fold for GEL_4 and ~2 fold for GEL_7, this has now been included in the main text (line 315).

1.12 Unclear from the manuscript body which diseases were included

The following has been added to the methods section to clarify the disease included in the 100,000 Genomes Project (line 521-524):

The rare disease cohort includes individuals with a wide array of diseases including neurodevelopmental disorders, cardiovascular disorders, renal and urinary tract disorders, ophthalmological disorders, tumour syndromes, ciliopathies and others. These are described in more detail in previous publications^{59,60}

1.13 Unclear what coverage was used in the initial cohorts

Apologies that this was not clear. The 100,000 Genomes Project individuals were whole genome sequenced at ~35X mean coverage. These details have been added to the Methods as well as a citation to a previous publication which has already described sequencing methods in detail. The initial DDD cohort was exome sequenced and subsequently the 9 potential hypermutated individuals and their parents were subsequently whole genome sequenced at a coverage of >30X mean coverage which is described in the Methods section and also now mentioned in the main text (see response to 1.7)

1.14 Supplemental Fig 3, 5, 6 does not have a caption

These have now been added to make the figures clearer.

1.15 Supplemental figures are out of order

These are now in order.

“Heterozygous PTVs in known somatic mutator genes do not always have a similar effect in the germline”. This may be a bit strong since only one gene (MBD4) was tested among n=13 paternal carriers of MBD4 PTVs

This has been reworded as follows to weaken the statement:

*Heterozygous PTVs in known somatic mutator genes **may** not always have a similar effect in the germline*

Aaron Quinlan (mentor) and Jason Kunisaki (graduate student)

Referee #2 (Remarks to the Author):

The manuscript by Kaplanis et al entitled “Genetic and pharmacological causes of germline hypermutation” is well written and straightforward to follow. The authors analyzed genome wide sequences from 21,879 families with rare genetic diseases and identified 12 hypermutated individuals with excessive numbers of de novo single nucleotide variants and small indels after correcting for paternal age and sequencing quality. This is a large data set and the study is straight forward. The authors do a nice job of phasing the variants and characterizing the mutation signatures. This is important to understand mechanism for the hypermutability phenotype. The authors also do a nice job of more precisely estimating the paternal age effect which appears to account for slightly less of the variance than the previous estimates based on a much smaller sample set. The results however are largely negative and obvious. Because the results are negative, they are largely reassuring that the absolute frequency of hypermutation is low. That risk may be slightly higher in consanguineous communities, but even there would be quite low. There is also largely a negative result to suggest that environmental carcinogens are a major contributor to hypermutation.

2.1 Of the 12 individuals who are hypermutated, six of them had fathers who received chemotherapy prior to conception. This is well known clinically that chemo is associated with increased frequency of mutations and is one of the reasons why we often bank gametes prior to chemotherapy.

We agree with the reviewer that the effect of chemotherapeutic treatment on mutation burden is well-known mainly from somatic, *in vitro* and animal studies. However, to our knowledge this is the first direct demonstration of the impact of chemotherapeutic treatment on germline *de novo* mutations in humans. Despite the fact that sperm banking is routinely being offered to the young men who undergo chemotherapy, natural conception post-treatment is favoured to IVF with banked samples as spermatogenesis tends to recover a few months after chemotherapy (Okada and Fujisawa 2019). In cases where chemotherapy perturbs sperm production, resulting in a high risk of oligozoospermia or azoospermia, banked samples are being used. Hence, we believe this study highlights the importance of considering using banked samples for family planning in men who undergo chemotherapy treatments.

2.2 There are two interesting cases of fathers with bilallelic recessive variants in DNA repair genes. One father is clinically affected with xeroderma pigmentosa due to homozygous nonsense variant in XPC associated with nucleotide excision

repair. It is not surprising that he would have gametes that are hypermutated. A second father has a rare homozygous missense variant in MPG. This is more interesting since MPG has not been described as a cancer susceptibility gene previously although the gene encodes a gene that is involved in the initiation of base excision repair pathway.

2.3 There are interesting confounds by indication for this analysis. Although several of the fathers received chemotherapy, this was also because they had cancer at a young age. I would be very careful to check that there are no germline cancer susceptibility genes although I appreciate that the authors think they did this. For example, the father of GEL_11 was diagnosed with three separate cancers at a young age. Osteosarcoma, lung cancer, and cancer of the intestinal tract is quite an unusual spectrum, especially for such a young individual. I would look more carefully at that father for some sort of heritable cancer syndrome or environmental exposure.

The father of GEL_11 initially had osteosarcoma and then the lung cancer recorded 2 years later is reported as a secondary cancer so would not be considered a separate event. The intestinal cancer code is not explicitly noted as a secondary cancer but it was recorded less than a year after the secondary lung cancer so it is likely this is also a metastatic event. Nevertheless, in addition to rare nonsynonymous variants in known DNA repair genes in the father for this individual, we also looked for rare (MAF <0.01) nonsynonymous variants in genes known to be associated with cancer (Martincorena et al. 2018) and did not identify any putative variants. Moreover if there is a heritable cancer syndrome in this patient then this might not necessarily be the source of hypermutation due to the signatures present. The largest signature contribution in the child is SBS31 which is associated with cisplatin use in somatic mutations suggesting this is the most likely cause of the hypermutation.

2.4 Somatic mutations in the child are rare in young children but increase with age. It would be helpful to list the age of the children in table 1.

This has now been added to the table.

2.5 For the GEL_7 child, it is not clear to me if the child has the TP53 variant observed to be mosaic in the mother. If so, the variant is germline and maybe a contributor to that hypermutation.

The TP53 variant is not observed in the child but could be present in the maternal germline and have contributed mutant protein to the zygote and could possibly be the contributor of the hypermutation. We have clarified this in the manuscript.

2.6 Because there can be technical issues with these variant calls, I recommend that the authors confirm the presence of a representative sample of the variants in the hypermutable cases. This is especially important for the cases of unknown source of hypermutation.

We agree with the reviewer that it is important to have high confidence that the called DNMs in the hypermutated individuals are true positives rather than artifactual calls.

There are seven lines of evidence that the DNM calls in the hypermutated individuals are not enriched for artefacts:

1. DDD_1 was originally detected due to an excess of DNMs in exome sequencing data prior to generating the WGS data. Thus providing two independent experiments validating hypermutation in this individual.
2. The signature analysis included 18 signatures of common sequencing artefacts and we did not observe these in any of the hypermutated individuals.
3. Other than the novel signature seen in DDD_1 and GEL_2, all other detected signatures have been previously validated as being due to exogenous or endogenous factors in somatic mutation studies
4. No evidence of clustering in the genome that might have suggested that a somatic loss of transmitted allele had been over-looked (Reviewer Figure 1)

Reviewer Figure 1: Distribution of DNMs across autosome for hypermutated GEL individuals. Grey bar is length of chromosome and colored dots are position of DNMs

5. The VAF distribution of DNMs in hypermutated individuals, excluding GEL_4 and GEL_7 who have post-zygotic hypermutation, is not significantly lower than DNMs in non-hypermutated individuals (Reviewer Figure 2). Indeed the median VAF of hypermutated DNMs is actually slightly higher (0.470 vs 0.465, $p = 0.0006$, Mann-Whitney test). This is not unexpected, if we assume a uniform rate of false positive calls across all individuals, the proportion of false positive calls in truly hypermutated individuals would be expected to be lower than in non-hypermutated individuals.

Reviewer Figure 2: VAF distribution of DNMs in hypermutated and non-hypermutated individuals

6. Sequence errors would typically be expected to occur on both haplotypes. The proportion of variants phasing robustly to a single haplotype in hypermutated individuals is not different from non-hypermutated individuals ($p=0.49$, Binomial test). Those variants that can be phased typically show a strong sex bias. If sequencing errors were being erroneously phased this would be expected to occur equally on maternal and paternal haplotypes.
7. Manual inspection of IGV plots for all DNMs in hypermutated individuals and 500 randomly selected DNMs across the non-hypermutated individuals, showed that hypermutated DNMs had a significantly lower likely false positive rate for both SNVs (0.8% hypermutated vs 2.7% non-hypermutated, $p = 1.46e-10e-5$, Binomial Test) and indels (0% hypermutated vs 3% non-hypermutated, $p = 0.0018$, Binomial Test).

There are significant DNA sample governance challenges to solve in order to access the DNA samples from GEL for additional validation experiments. Solving these challenges has the potential to introduce substantial delays without guarantee of a successful outcome. We hope that the reviewer agrees that the above evidence (in combination with our >10 year track record of generating high quality DNM callsets) is sufficient to have high confidence in these data.

2.7 A minor point: I would add a few fields to table 1: the type of chemotherapy received by the father if it is known and the type of cancer in the father if it is known.

These have now been included in Table 1

2.8 Please provide the list of DNA repair genes used in this analysis and how did you select them?

As detailed in the methods, we compiled a list of DNA repair genes which were taken from an updated version of the table in Lange et al, Nature Reviews Cancer 2011 available on the website <https://www.mdanderson.org/documents/Labs/Wood->

Laboratory/human-dna-repair-genes.html (accessed March 2020). The exact genes have also been added as Supplemental Table 3.

2.9 I would replace pharmacological in the title to specify that the drugs they examined were chemotherapeutic agents.

Thank you for this suggestion, we have changed the title to reflect this.

Referee #3 (Remarks to the Author):

Summary of the study and key findings:

Kaplanis and colleagues have conducted a very interesting and important study into the causes of germline hypermutation and identified novel contributing factors, including paternal exposure to chemotherapy prior to conception and paternal genetic variants in DNA repair genes. This study is much larger than previous analyses, utilizing two cohorts of parent-offspring trios totaling ~20,000 families with children that have developmental disorders or rare diseases.

The authors confirm, as shown previously in smaller studies, that the majority of germline mutations are explained by parental age, particularly of the father. They were able to phase ~25% (225,854/921,433) de novo mutations (DNMs) and found that 77% of these were paternal in origin. The majority of both maternal and paternal mutations can be explained by COSMIC signatures SBS1 and 5.

The authors went on to identify 12 individuals hypermutated in the germline (after accounting for parental age), harbouring ~2-7-fold more DNMs than the median number of DNMs/individual in the 100kGP cohort. They assessed the mutational spectra of the DNMs in each individual and used SigProfiler to extract mutational signatures. These mutational signatures were then used to identify the potential causative mechanism(s) for the hypermutation.

The mutational signatures in two patients indicated a contribution of DNA repair deficiency. In both cases, the fathers were found to carry homozygous nonsynonymous variants in DNA repair genes. In one case (GEL_1) the father had a rare nonsense variant in XPC, a key component in the nucleotide excision repair pathway, and the child had mutations which match signature SBS8, associated with NER-deficiency. While XPC deficiency is known to increase the risk of developing cancer, the link to human germline mutagenesis is novel. In the second case (GEL_3), a contribution of SBS26 is observed, which is associated with defective mismatch repair. The father has a missense variant (A135T) in the gene MPG, which codes for a glycosylase involved in the repair of alkylated and deaminated purines. Using in vitro functional assays, the authors show that this variant changes the activity of MPG, but further work would be required to establish a link between the A135T variant and SBS26.

Significantly, the authors also establish a link between germline mutations and paternal exposure to chemotherapy. SBS31, a signature associated with platinum-drug exposure, contributed to the DNM mutational spectra of three children. The fathers of these children had a history of cancer and chemotherapy prior to conception, implicating platinum-based treatment as a source of germline mutagenesis. Two other children shared paternally-biased DNM mutational spectra, from which the authors extracted a novel mutational signature named SBSHYP. As both fathers received treatment with nitrogen mustard alkylating agent chemotherapy prior to conception, these agents are suspected to be the cause of the novel signature.

Finally, the high level of DNMs in two further individuals was found to be caused by post-zygotic hypermutation, and consisted of SBS1 and 5.

General comments on the study and manuscript:

This is a well written article, clearly presented and accessible. The design of the study, the methods used and the interpretation of the results appear to be sound. Analyses were well thought-out and thorough, documented in clear Figures and Tables. Statistical tests and probability values are described for each analysis. Conclusions drawn from the results are valid and not overstated. The reference of previous literature is appropriate and relevant.

The work presented here builds upon previous studies by utilising a larger dataset of parent-offspring trios. This study confirms previous findings (e.g. the proportion of DNMs attributable to the father and mother) but also provides important, new insight by identifying causative elements for the mutational signatures found in germline hypermutated individuals (e.g. paternal chemotherapy exposure), which has not been shown previously. The authors also highlight that germline hypermutation is ultimately rare. The findings in this study will certainly be of interest to a diverse audience, spanning the fields of human genetics, DNA repair and genetic toxicology and will stimulate further research.

Specific comments for discussion or improvement:

3.1 . Fig 1b Please lighten the purple colour as it is difficult to read black numbers on dark purple.

We have lightened the colour to make this easier to read.

3.2 Fig 2a, It may be useful to some readers for the classification of each of these signatures to be briefly labeled on the Figure or in the legend (ie SBS31: platinum drug; SBS26: MMR-deficiency).

We have added brief descriptions of these signatures in the legend.

3.3. Supplemental Figure 13 indicates that one would expect some contribution of SBS5 in every individual, as this signature is detected in all of the controls, as well as the non-hypermutated individuals with parents who received

chemotherapy. Therefore, could the authors explain, or discuss, why there is no contribution of SBS5 in GEL_2 or GEL_11?

We agree with the reviewer that SBS5 is expected to be present in all cell types. However, in GEL_2 and GEL_11, due to the fact that a high proportion of mutations were attributed to the novel signature, SBSHYP, detection of the ubiquitous signature, SBS5, was limited using SigProfiler. Hierarchical Dirichlet Process (HDP) is a nonparametric Bayesian approach for mutational signature analysis (Teh et al. 2006). Both SigProfiler and HDP are widely used for signature extraction. While SigProfiler, which is a non-negative matrix factorization, has a high specificity (more conservative method) in extracting signatures it has a lower sensitivity to signatures that are present at a lower level. However, HDP analyses the significant differences in signature prevalence across samples and groups in the hierarchy using a formal probabilistic model with credibility intervals for every parameter. Hence, it has a higher sensitivity in distinguishing signatures that are less prevalent. To address the reviewer question, we ran HDP on all the samples in Supplementary Figure 11 and deconvoluted signatures. For this step the list of candidate signatures were restricted to SBS1, SBS5, SBS8, SBS16, SBS24, SBS26, and SBS31 (informed from the SigProfiler run). Signatures were then refitted using SigFit, in each case according to HDP signatures (and inferred reference signatures) present in each sample, with SBS1 and SBS5 always included. We observed that the SBSHYP signature extracted from SigProfiler and that extracted from HDP are very similar (cosine similarity 0.997). Also, SBS5 was extracted from GEL_2 and GEL_11 mutations (1.5%, 5 mutations and 7.6%, 8 mutations respectively) (**Reviewer Figure 3**). Due to the greater specificity of SigProfiler, we prefer to use SigProfiler rather than HDP in our primary analyses.

Reviewer Figure 3 | a. Triplet mutational profile of SBSHYP using HDP and SigProfiler **b.** Correlation between mutations assigned to each signature using SigProfiler and HDP method **c.** Mutations extracted in hypermutators using HDP method

3.4 In the Results section “Paternal defects in DNA repair”, the authors show that GEL_3 harbours mutations that map to SBS26, which is linked to defective MMR, and the father has a variant in the gene MPG. Could the authors comment on other MPG variants present in humans? For example, I found at least one study describing human variants in MPG: <https://doi.org/10.1074/jbc.M114.627000>. doi.org Some in vitro work was done by the authors to show that the activity of MPG may be affected by the A135T variant. The variant had decreased activity for one adduct and increased activity for another. Could the authors put these results into context, as it is not clear how significant a 2-fold decrease or increase would be. Have any other variants for MPG been assessed in the in vitro assay used by the authors, and if so, how did those results compare with the results presented for the A135T variant?

To put our findings in a broader context we have now generated an additional supplementary table, with references, that summarises the different MPG variants that have been functionally characterised in different studies, including in the

Adhikari et al. 2015 paper highlighted by the reviewer (Supplemental Table 5). These studies collectively describe 15 variants, all but one of which exhibited a difference in the single turnover rate compared to wildtype for an ethenoA and/or hypoxanthine substrate. Only three of these variants are represented in gnomAD, including the A135T variant characterised in our study. Of all the variants characterised, the A135T variant causes the largest increase in the single turnover rate constant for removal of ethenoA. The only variant that causes a comparably large increase in ethenoA excision is N169S, which exhibits a mutator phenotype when expressed in yeast cells.

3.5 In the Results section “Parental treatment with chemotherapy prior to conception”, it would be useful to briefly describe the type of DNA damage and mutations caused by cisplatin, rather than simply referring to SBS31 (especially since SBS31 is not illustrated anywhere in the manuscript). Further, could the authors comment on whether there is any evidence in the literature that platinum drugs induce germline mutation in animal models?

We have added the following to the text (line 248):

Platinum-based drugs damage DNA by causing covalent adducts. Cisplatin mainly reacts with purine bases, forming intrastrand crosslinks which can be repaired by NER or bypassed by translesion synthesis which may, in turn, induce single base substitutions(Boot et al. 2018).

The only study we are aware of that has studied the effect of cisplatin on germline mutation rates in mammals is the study in 2000 by Barber et al(Barber et al. 2000) that showed no effect on ESTR mutation rate with cisplatin. However, ESTRs mutate by a different mechanism than SNVs and so we didn't think this study was sufficiently relevant to cite here.

3.6 Supplemental Figure 2: The y-axis labeling needs to be fixed in (b).

This has been fixed.

3.7 Supplemental Figure 5: In the lefthand panel, many of the graphs have bars that are higher than the y-axis. Please check and amend. For example, see e) and h).

These have been amended.

3.8 Supplemental Figure 10: Again, check the height of the y-axis, as some bars overtake the axis label. For example, see MBD4_9 and MBD4_13.

These have been amended.

3.9 Supplemental Figure 12: Please improve the resolution of this figure as it is almost impossible to read the text when zoomed in.

We have now addressed this issue.

References

- Barber, R., M. Plumb, A. G. Smith, C. E. Cesar, E. Boulton, A. J. Jeffreys, and Y. E. Dubrova. 2000. "No Correlation between Germline Mutation at Repeat DNA and Meiotic Crossover in Male Mice Exposed to X-Rays or Cisplatin." *Mutation Research* 457 (1-2): 79–91.
- Boot, Arnoud, Mi Ni Huang, Alvin W. T. Ng, Szu-Chi Ho, Jing Quan Lim, Yoshiiku Kawakami, Kazuaki Chayama, Bin Tean Teh, Hidewaki Nakagawa, and Steven G. Rozen. 2018. "In-Depth Characterization of the Cisplatin Mutational Signature in Human Cell Lines and in Esophageal and Liver Tumors." *Genome Research* 28 (5): 654–65.
- Martincorena, Iñigo, Keiran M. Raine, Moritz Gerstung, Kevin J. Dawson, Kerstin Haase, Peter Van Loo, Helen Davies, Michael R. Stratton, and Peter J. Campbell. 2018. "Universal Patterns of Selection in Cancer and Somatic Tissues." *Cell* 173 (7): 1823.
- Okada, Keisuke, and Masato Fujisawa. 2019. "Recovery of Spermatogenesis Following Cancer Treatment with Cytotoxic Chemotherapy and Radiotherapy." *The World Journal of Men's Health* 37 (2): 166–74.
- Pich, Oriol, Ferran Muiños, Martijn Paul Lolkema, Neeltje Steeghs, Abel Gonzalez-Perez, and Nuria Lopez-Bigas. 2019. "The Mutational Footprints of Cancer Therapies." *Nature Genetics* 51 (12): 1732–40.
- Teh, Yee Whye, Michael I. Jordan, Matthew J. Beal, and David M. Blei. 2006. "Hierarchical Dirichlet Processes." *Journal of the American Statistical Association* 101 (476): 1566–81.

Reviewer Reports on the First Revision:

Referees' comments:

Referee #1 made no comments to the authors

Referee #2 (Remarks to the Author):

The authors have adequately addressed my previous comments, and the revised manuscript is acceptable for publication.

Referee #3 (Remarks to the Author):

The authors have done a thorough job addressing all of the reviewers' comments and concerns. The manuscript has been sufficiently revised and I would now personally recommend it for publication.